# Fermented foods affect the seasonal stability of gut bacteria in an Indian rural population

Kumaraswamy Jeyaram [1,2] ✉, Leo Lahti[2,3,4], Sebastian Tims[2,5], Hans G. H. J. Heilig [2], Antonie H. van Gelder[2], Willem M. de Vos [2,3], Hauke Smidt [2] & Erwin G. Zoetendal [2]

The effect of fermented foods on healthy human gut microbiota structure and function, particularly its seasonal preference and frequent long-term consumption, has been largely uncharacterised. Here, we assess the gut microbiota and metabolite composition of 78 healthy Indian agrarian individuals who differ in the intake of fermented milk and soybean products by seasonal sampling during hot-humid summer, autumn and dry winter. Here we show that, seasonal shifts between the *Prevotella*- and *Bifidobacterium/Ruminococcus*-driven community types, or ecological states, and associated fatty acid derivatives, with a bimodal change in Bacteroidota community structure during summer, particularly in fermented milk consumers. Our results associate long-term fermented food consumption with reduced gut microbiota diversity and bacterial load. We identify taxonomic groups that drive the seasonal fluctuation and associated shifts between the two ecological states in gut microbiota. This understanding may pave the way towards developing strategies to sustain a healthy and resilient gut microbiota through dietary interventions.

Food fermentation can be considered an upstream extension of food digestion, in which the complex food matrix is transformed by a microbial process that may enhance its nutritional value. About one-third of our food is fermented before consumption[1]. Several sources of ancient literature and indigenous traditional knowledge worldwide have linked fermented foods with health benefits[2,3]. The prolonged life of Bulgarians has been related to fermented milk consumption[4]. Nowadays, many of these artisanal food products have been scaled to industrialised processes and received attention for their health-promoting potential. Most health claims are on fermented foods and associated bacteria on the host metabolism and diseases. There are no validated health claims on fermented foods, except the claimed effect of yoghurt for lactose digestion in individuals with lactose maldigestion validated by the European Food Safety Authority (EFSA)[5]. However, the impact of fermented foods on human gut microbiota composition has not yet been clearly established. The most abundant human gut bacteria include the phyla Bacillota, Bacteroidota, Actinomycetota, Pseudomonadota, and Verrucomicrobiota[6], whereas food fermentation mainly involves lactic acid bacteria and *Bacillus* spp. of the phylum Bacillota, in addition to yeasts and fungi. Understanding the impact of fermented foods on gut microbiota composition may allow us to innovate novel functional foods.

In recent years, the gut microbiota has been recognised for its contribution to healthy immune homeostasis, and differences in gut microbiota composition have been linked with many health issues[7]. Evidence shows that diet plays a dominant role in impacting the gut microbiota composition[8], whereas genetics is likely not a valid driver. The overall inter-individual difference in human gut microbiota composition is explained mainly by the dominance of *Prevotellaceae* and *Bacteroidaceae* (two prominent families of the phylum Bacteroidota).

[1]Biotechnology Research and Innovation Council - Institute of Bioresources and Sustainable Development (BRIC-IBSD), Regional Centre, Tadong, Gangtok 737102 Sikkim, India. [2]Laboratory of Microbiology, Wageningen University & Research, 6708 WE Wageningen, The Netherlands. [3]Human Microbiome Research Program, Faculty of Medicine, University of Helsinki, 00014 Helsinki, Finland. [4]Department of Computing, University of Turku, FI-20014 Turku, Finland. [5]Danone Nutricia Research, 3584 CT Utrecht, The Netherlands. ✉e-mail: jeyaram.ibsd@nic.in

Based on the distinct host-microbial symbiotic states, gut microbiota broadly group into *Bacteroides* enterotypes 1 and 2, and *Prevotella* and *Ruminococcaceae* enterotypes[9–11]. *Prevotellaceae* is more common in the human population that follows a traditional lifestyle, whereas *Bacteroidaceae* dominates in the populations exposed to modern industrialised lifestyles[12]. The two states of this bimodal distribution (alternative stable states) have been suggested to be due to long-term dietary habits[13]. David et al. [14] demonstrated that even a short-term change in an extreme diet could rapidly and reproducibly alter the gut microbiota, particularly the prevalence of *Prevotella*. The level of resilience is individual-specific; some return to their original state after perturbation, whereas in others, a new stable ecological state can appear[8,15,16]. Another study with natural diets that differ in fibre content showed that dietary changes lead to bacterial activity changes, particularly affecting butyrate producers in a two weeks intervention[17]. In the case of Hadza hunter gatherers, changes in the gut microbiota populations are cyclic (annual) due to the seasonal weather and dietary changes[12]. These seasonal cyclic changes in human gut microbiota are mostly observed in people living in traditional societies[18]. Applying a unified definition for the healthy configuration of gut microbiota is challenging[19].

Several studies have shown that consuming probiotic supplements or fermented foods with live bacteria can beneficially impact health in diseased conditions. Examples include a symbiotic preparation with *Lactiplantibacillus plantarum* and fructooligosaccharide prevented sepsis among infants in rural India[20], and *Limosilactobacillus reuteri* DSM17938 was effective in breastfed infants with colic[21]. With the support of few human clinical studies, the International Scientific Association for Probiotics and Prebiotics (ISAPP) consensus statement on fermented foods[22] stated evidence of health benefits in yoghurt consumption with a reduction in type-2 diabetes mellitus and cardiovascular diseases. The ISAAP-sponsored National Health and Nutrition Examination Survey (NHANES) analysis witnessed positive health outcomes associated with the intake of foods with live microbes, including fermented foods, as it was associated with a reduction in systolic blood pressure and an increase in HDL, which are known to decrease the risk of cardiovascular diseases[23]. Studies to determine the impact of fermented foods with live bacteria on the healthy gut bacterial ecosystem mainly address two issues: the fate of the ingested bacteria and the changes in gut microbiota composition[24]. Based on these transient bacteria's sheltering ability in gut microbiota, Zhang et al. [15] categorised the gut microbiota into colonization resistance (its ability to prevent colonization by allochthonous or exogenous bacteria by eliminating them immediately) and colonization permissive (shelter the exogenous strain over 24–48 h). In most cases, ingested bacteria are numerically a minority and rarely detected one week after consumption[15,25,26]. Several studies show no significant change in the dominant gut microbiota of healthy adults following consumption of fermented foods with probiotic bacteria[24]. However, a large-scale metagenome-assembled genomes (MAGs) analysis provides unprecedented evidence that fermented foods can indeed be the primary source of lactic acid bacteria in the human gut[27]. Moreover, Ferrario et al. [28] noticed a shift in the gut microbiota composition mostly by stimulating butyrate producers.

Earlier studies on the impact of fermented foods on healthy gut microbiota were designed using short intervention (3-7 weeks) with few bacterial strains. We hypothesised that long-term consumption (>10 years) of naturally fermented foods would impact the gut microbiota composition. Moreover, the seasonal volatility noticed in the prevalence of *Prevotellaceae*[12], the dominant gut bacterial family of the Indian population, and the indigenous traditional knowledge of preferring different fermented foods during summer and winter in India[29] drive us to study the impact of fermented foods on seasonal gut microbiota changes. We selected two of the most commonly consumed fermented foods with live bacteria: a fermented milk product *Dahi* (acidic fermentation by lactic acid bacteria)[30] preferred more frequently during summer with a traditional belief of cooling effect, and a fermented soybean product *Hawaijar* (alkaline fermentation by *Bacillus*)[31] preferred more regularly during winter with a traditional belief of having winter wellness. Based on self-reported consumption frequency in the intake of the above two fermented foods, a genetically and culturally homogenous endogamous agrarian population of the Meitei community in Manipur, India, living in a geographically isolated valley region surrounded by mountainous forest terrain with minimum contact with other social groups were targeted for this study. Our study intends to understand the impact of long-term consumption of fermented foods on seasonal changes in gut microbiota composition on this isolated rural Indian community by analysing the faecal samples, as the faecal microbiota are commonly used as a marker for the colon microbiota populations.

Here, we show the seasonal instability in the human gut microbiota due to long-term fermented food consumption and the bacterial drivers for this ecological instability.

## Results

### Gut bacterial composition formed a bimodal distribution in the Indian population

Our survey identified 78 healthy subjects, who differed in the intake of fermented foods categorised into four diet groups, Group-A: never consumed *Hawaijar* and *Dahi* (n = 20, control); Group-B: consumed both *Hawaijar* and *Dahi* (n = 21); Group-C: consumed *Hawaijar*, not *Dahi* (n = 23); and Group-D: consumed *Dahi*, not *Hawaijar* (n = 14). The study groups were balanced in age, sex, body mass index (BMI), nature of birth, dietary and lifestyle habits (Suppl. Table S1). We collected the faecal samples over three-time intervals (hot-humid summer, autumn, and dry-winter) and determined the bacterial composition using Human Intestinal Tract Chip[6] (HITChip; a 16S rRNA gene-based phylogenetic microarray targeting the gut microbiota) supported with taxa-specific qPCR assays, determined the changes in faecal metabolites profiles by liquid chromatography-high resolution mass spectrometry (LC-HRMS) and quantified the short-chain fatty acids (SCFA) by high-pressure liquid chromatography (HPLC).

Faecal bacterial community composition-based grouping of the Indian study population by Pearson product-moment correlation analysis of HITChip probe signals generated two distinct clusters (Fig. 1A, Suppl. Fig. S1). Further, PCoA plotting (Fig. 1B) of faecal bacterial compositional abundance at genus-level by Bray-Curtis dissimilarity visualised the detected two clusters. Redundancy analysis by correlating metadata showed that these two clusters did not separate based on the categorised study groups, seasons, short-term (2 days' diet recall), long-term dietary habits, clan, age, sex, BMI, or lifestyle habits. We observed a shift of samples in 17 subjects between the two clusters during seasonal sampling, collected over three-time points (Fig. 1C). The redundancy analysis could also not associate this seasonal shift between the two clusters with short-term dietary habits (2 days' diet recall). Our analysis visualised *Prevotella melaninogenica* et rel. as the key driver for separating these two clusters ($p = 2.5 \times 10^{-21}$, Wilcoxon test, BH corrected, Suppl. Fig. S2A), the cluster with a higher relative abundance of *Prevotella* named as *Prevotella*-driven cluster (Cluster-P), whereas the cluster with a higher relative abundance of *Bifidobacterium* and *Ruminococcus* named as *Bifidobacterium/Ruminococcus*-driven cluster (Cluster-B/R). When calculating the taxon-specific abundances by multiplying the relative abundances by the total bacteria 16S rRNA gene copy numbers, the absolute abundance of *Prevotella* was significantly different between the two clusters ($9.93 \pm 0.11$ in Cluster-P and $7.33 \pm 0.41$ in Cluster-B/R; median ± s.e.m., log10 16S rRNA gene copies g$^{-1}$ of the wet faecal sample). However, *Bifidobacterium* and *Ruminococcus* did not show absolute abundance differences between the clusters (Suppl. Table S2), indicating that *Prevotella* is the driving taxon for the formation of the two clusters.

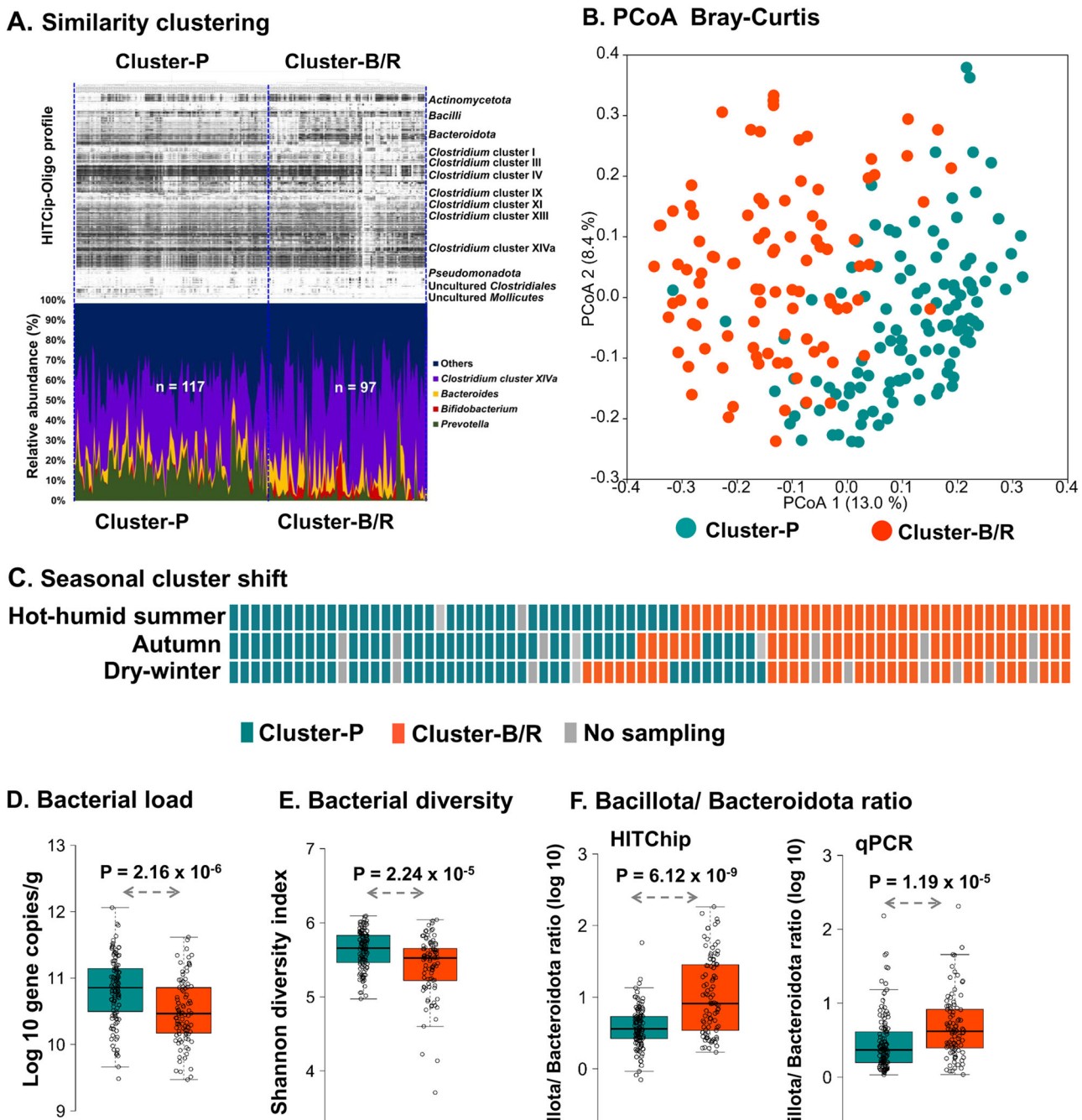

**Fig. 1 | Gut microbiota composition based grouping of the Indian study population. A** Two clusters revealed through hierarchically clustered HITChip oligo profile (*n* = 214, biological replicates, complete hierarchical clustering of the log10 transformed HITChip probe signal oligo profile with Pearson similarity, *see Methods*) named here as *Prevotella*-driven Cluster-P and *Bifidobacterium/Ruminococcus*-driven Cluster-B/R. The intensity of the heat map relates to the relative abundance (%) of the major bacterial phylum, as shown in the legend. The stacked chart shows the relative abundance of the major discriminating taxa (*p* < 0.01, two-sided Wilcoxon test, corrected by the Benjamini-Hochberg method) between two clusters. **B** PCoA for the genus level community composition (Bray-Curtis dissimilarity for the relative abundance) shows the separation of the two clusters. **C** The diagram shows the shift of gut bacterial clusters over three-time points (seasons) sampling in the individuals of the study population. Cluster-P, blue-green and

Cluster-B/R, orange-red. **D–F** The Box and whisker plots show the difference in the total bacterial load (total bacteria-specific qPCR assay, expressed as log 10 gene copies g⁻¹ of the wet faecal sample) (**D**), difference in bacterial diversity (generated from the HITChip probe profile) (**E**) and difference in the Bacillota/Bacteroidota ratio from HITChip probe signals and the taxa-specific qPCR assay (**F**) between the Cluster-P and Cluster-B/R. The Box and whisker plots display median (middle line), box ranges from 25 to 75 percentile with Tukey whiskers and outlier (more than 1.5 IQR). The statistical significance of the difference between the biological replicates of Cluter-P (*n* = 117) and Cluster-B/R (*n* = 97) was calculated by a two-sided Wilcoxon rank sum test with Bonferroni correction, and the *p*-values are indicated in the figure panel (**D–F**). The box colour of Cluster-P is blue-green, and Cluster-B/R is orange-red. Source data are provided as a Source Data file.

We did a confounder analysis using Linear Regression (SPSS Statistics) to find out other significant factors driving these two bacterial community clusters by checking the variability in the recruited subjects (age, sex, BMI), habitual diet and lifestyles, and 48 h diet recall data retrieved from a questionnaire used during sample collection. The variability in diet and lifestyle habits of individual subjects recruited for the study is provided as Supplementary Data. The confounder analysis resulted in a significant negative association of BMI with *Prevotella* (standardized regression coefficient $\beta = -0.287$, $p = 0.009$). BMI showed a subtle but significant difference between the two clusters, Cluster-P with a median BMI of 23.7 and Cluster-B/R with a median BMI of 25.7 ($p = 0.0021$, T-test, two-tailed, unequal variances). In addition, we observed a significant negative association of smoking with the *Clostridium* cluster-XIVa ($\beta = -0.381$, $p = 0.0047$) and the long-term-dietary habit of duck meat consumption with *Bifidobacterium* ($\beta = -0.402$, $p = 0.009$). However, smoking and duck consumption did not differ significantly between Cluster-P and Cluster-B/R.

The total bacterial load analysed by bacteria-specific qPCR assay (Fig. 1D) and the Shannon diversity index generated from HITChip analysis (Fig. 1E) were significantly different between these two clusters with a lower bacterial load and diversity in the Cluster-B/R ($p < 0.001$, Wilcoxon test, BH corrected). An absolute load of both Bacteroidota and Bacillota calculated by phylum-specific qPCR assays were also significantly lower in Cluster-B/R than in Cluster-P. However, the Bacteroidota load was much lower than the Bacillota load in Cluster-B/R (Cluster-P: Bacteroidota $10.91 \pm 0.08$, Bacillota $11.09 \pm 0.05$; Cluster-B/R: Bacteroidota $10.19 \pm 0.09$, Bacillota $10.68 \pm 0.06$, median $\pm$ s.e.m log10 of 16S rRNA gene copies g$^{-1}$ of wet faeces), reflected with a higher Bacillota/Bacteroidota ratio in the Cluster-B/R, and the Bacillota/Bacteroidota ratio calculated from HITChip probe signal data also supported the higher Bacillota/Bacteroidota ratio in the Cluster-B/R (Fig. 1F) ($p < 0.001$, Wilcoxon test, BH corrected). Further, genus-specific quantification by qPCR showed a significant difference in *Prevotella* load between the two clusters, without a difference in *Bacteroides* loads, indicating the difference in Bacillota/Bacteroidota ratio is driven by *Prevotella*.

#### Faecal metabolite composition supported the bimodality

To look at the difference between these two ecological states (Cluster-P and Cluster-B/R, defined based on the gut microbiota community structure, Fig. 1) from a metabolic point of view, we analysed the faecal extracts from 71 subjects at baseline (summer) covering 41 samples of Cluster-P and 30 samples of Cluster-B/R by LC-HRMS analysis. PCA analysis of the overall faecal metabolite profile data of samples sustained the separation of two clusters (PERMANOVA: $F = 5.7$, $p = 0.0001$), indicating an overall difference in the metabolism (Fig. 2A). LC-HRMS detected 871 metabolites in the faecal extracts, in which 59 compounds (6.8%) significantly ($p < 0.05$, two-tailed Wilcoxon test, Benjamini-Hochberg method corrected) differed between the two clusters. We assigned the identity of 23 significantly differing compounds between the two clusters by MS/MS spectral analysis by comparing it with the mzCloud library and listed in Supplementary Table S3. The hierarchically clustered heat map (Fig. 2B) shows the significantly differed metabolites (-2 log2-fold change, $p < 0.05$, two-tailed Wilcoxon test, Benjamini-Hochberg method corrected) between the clusters. Most of the differed compounds were at a higher level in the *Prevotella*-driven Cluster-P, whereas the *Bifidobacterium/Ruminococcus*-driven Cluster-B/R showed a higher level of C1871 (Linagliptin), C1166 (Ciprofibrate) and C1649 [Bis(2,3-dihydroxypropyl) 2-hydroxy-1,3-propanediyl bis [hydrogen (phosphate)]. Random Forest analysis predicted two long-chain fatty acid derivatives C1767 (a derivative of pentadecanoate) and C1806 (2-hexadecanoyl-sn-glycero-3-phosphoethanolamine), as the key differentiating metabolites ($\sim 20$ Mean decrease accuracy) between the two clusters (Suppl. Fig. S2B). Among the short-chain fatty acids quantified by HPLC, Cluster-B/R showed a

higher butyrate production ($p = 0.0013$, two-sided Wilcoxon test) (Fig. 2C), without any significant difference in acetate and propionate production between the Cluster-P and Cluster-B/R (Source Data). The confounder analysis with dietary habits resulted in a significant positive association of butyrate with pork consumption ($\beta = +0.255$, $p = 0.013$). However, pork consumption did not differ significantly between the Cluster-P and Cluster-B/R. The above analysis on gut microbiota and faecal metabolite profiling showed a bimodal distribution (two ecological states) of gut microbiota with different metabolisms in the Indian study population.

#### Comparison with European gut bacteria supports bimodality across the population

We compared the bimodal distribution observed in the Indian study population with a set of European individuals ($n = 78$; data from the HITChip database[32]) matched for age, BMI and gender. PCA using the gut bacterial compositional abundance of the most prevalent genus-like taxa (67 taxa, detected in >20% of the samples with above 2.8 log HITChip probe signal) showed two modes (Fig. 3A), separated mainly based on the *Prevotella* status in relation to PCA component-1. Multimodality analysis viewed a bimodality in *Prevotella melaninogenica* et rel., *Prevotella oralis* et rel., *Prevotella tannerae* et rel., *Ruminococcus bromii* et rel., and *Streptococcus mitis* et rel. in both (India and Europe). Among the bimodality taxa, only *Prevotella* differed significantly in their abundance between Indian and European subjects. The difference in gut bacterial composition between the two populations was mostly explained by a higher relative abundance of *Prevotella melaninogenica* et rel., *Catenibacterium mitsuokai* et rel., *Ligilactobacillus salivarius* et rel., *Megamonas hypermegale* et rel., *Veillonella* and *Megasphaera elsdeni* et rel. in Indian population, whereas European with a higher level of *Bifidobacterium*, *Eubacterium hallii* et rel., *Clostridium stercorarium* et rel. and *Anaerostipes caccae* et rel. ($p < 0.05$, Wilcoxon test, matched for age, BMI and gender, >2 absolute fold change, FDR < 0.25, BH corrected) (Fig. 3B, Suppl. Fig. S3). Overall, this comparative analysis indicated that the ecological state of European gut bacteria was towards Cluster-B/R, among the two ecological states visualised in the Indian study population.

#### Drivers of bimodal distribution in faecal bacteria

We generated a correlation co-occurrence network plot to understand the potential drivers of this bimodal distribution of faecal bacteria in the Indian study population. Our results showed a unique co-exclusion network between two modularity groups, namely modularity group-I (MG-I) with members of Bacteroidota and modularity group-II (MG-II) with members of *Clostridium* cluster XIVa (*Lachnospiraceae*) (Fig. 4). Similar significant co-exclusion of *Prevotella* with the members of *Clostridium* cluster XIVa (*Anaerostipes caccae* et rel., *Coprococcus eutactus* et rel, *Eubacterium hallii* et rel.) and *Bifidobacterium* was also visible after combining both Indian and European study populations (Suppl. Fig. S4). It is interesting to see low *Prevotella* and high *Bifidobacterium* in the European population and the other way around in the Indian population (Fig. 3B), which support the proposed drivers of the bimodal distribution observed in the study.

#### Fermented foods affect Bacteroidota stability during summer

To investigate the impact of the long-term dietary habit of fermented milk (*Dahi*) and fermented soybean (*Hawaijar*) consumption, we compared the faecal bacterial composition between the four diet groups. We observed a seasonal fluctuation of Bacteroidota (particularly in summer) reflected by a variation in the Bacillota/Bacteroidota ratio in the fermented foods consuming groups in the Indian study population without any change in the total bacterial load and bacterial diversity over the three seasons (Fig. 5). Both HITChip data and phylum-specific qPCR assay supported this seasonal fluctuation with a higher Bacillota/Bacteroidota ratio in the fermented food consumers,

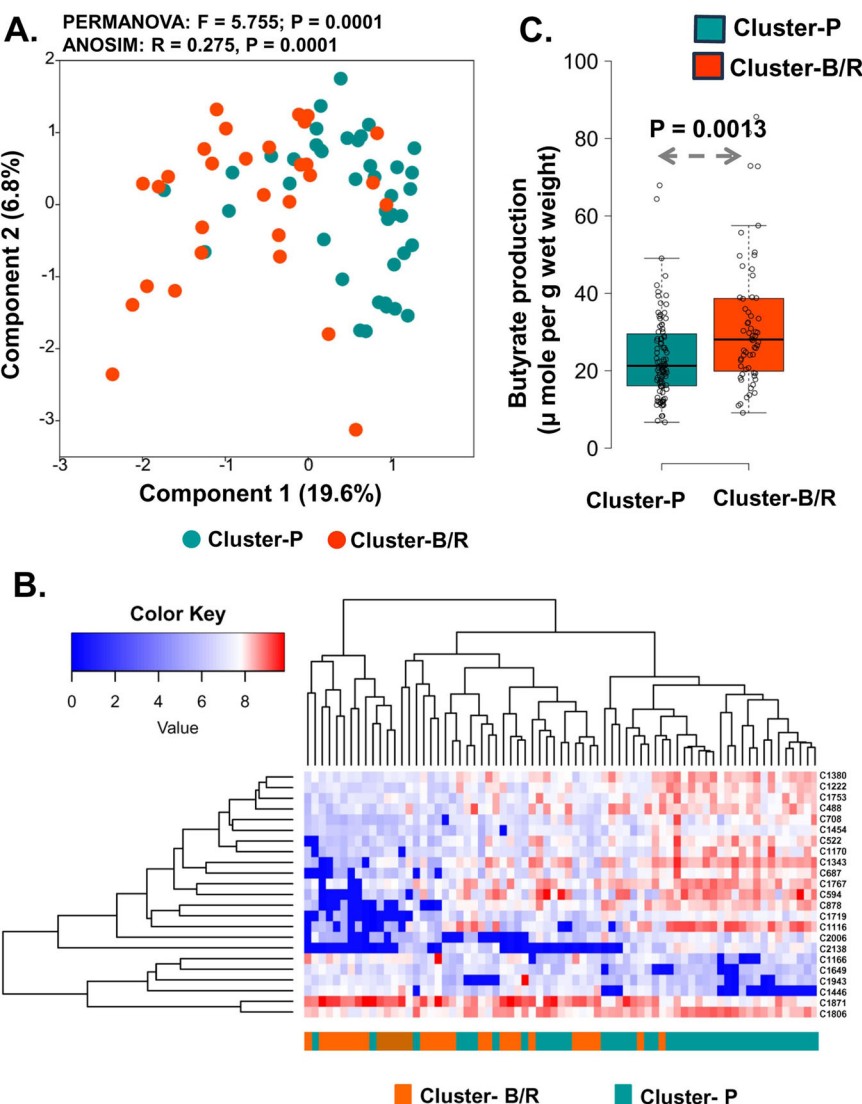

**Fig. 2 | Comparison of the faecal metabolite profiles of the two gut microbiota clusters in the Indian study population. A** PCA ordination of biological replicate samples ($n = 71$), at baseline (summer) based on faecal metabolites LC-HRMS data (log-transformed) support the two gut microbiota-based clusters, *Prevotella*-driven Cluster-P and *Bifidobacterium/Ruminococcus*-driven Cluster-B/R. **B** The hierarchically clustered heat map shows the clustering of 23 faecal metabolites significantly different between the two clusters (~2 log2-fold change, $p < 0.05$, two-sided Wilcoxon test, Benjamini-Hochberg method corrected). A colour gradient of blue and red was used to visualise the difference in the level of the metabolites.

**C** The Box and whisker plot shows the difference in butyrate production (μ mole per g wet weight) between Cluster-P and Cluster-B/R. The Box and whisker plots display median (middle line), box ranges from 25 to 75 percentile with Tukey whiskers and outlier (more than 1.5 IQR). The statistical significance of the difference between the biological replicates of Cluter-P ($n = 96$) and Cluster-B/R ($n = 62$) was calculated with a two-sided Wilcoxon rank sum test, and the *p*-values are indicated in the figure panel (**C**). The colour of Cluster-P is blue-green, and Cluster-B/R is orange-red in all the figure panels. Source data are provided as a Source Data file.

---

particularly in summer ($p < 0.05$, two-sided Wilcoxon test). Surprisingly, the Bacillota/Bacteroidota ratio in control Group-A was highly steady over the three seasons ($p > 0.9$). This seasonal fluctuation of Bacteroidota supported the seasonal shift of samples between the two clusters of ecological states over a study period of 6 months (Fig. 1C).

**Fermented food consumption in the long-term change the gut bacterial ecological state**

We observed a significant change in the gut bacterial load and Shannon diversity (Fig. 6) in the fermented foods consuming groups in comparison to the control Group-A, without any change over the three seasons (Fig. 5B, C). The qPCR assay and HITChip data showed a lower faecal bacterial load ($p < 0.001$, two-sided Wilcoxon test), and a lower Shannon diversity, respectively, in the fermented food consumer groups (Fig. 6A, B). Like the total bacterial load, the fermented food

consumer groups showed a significantly lower absolute abundance of Bacteroidota and Bacillota, with more effect on Bacteroidota (Suppl. Fig. S5). The above results of the high Bacillota/Bacteroidota ratio, low bacterial diversity, and low bacterial load suggested possible gut bacterial composition changes towards *Bifidobacterium/Ruminococcus*-driven Cluster-B/R in the fermented food consumers. We quantified the *Prevotella* and *Bifidobacterium* loads (as an indicator for Cluster-P and Cluster-B/R, respectively) to support the change in the gut bacterial ecological states (bimodality) in the diet groups. This taxa-specific qPCR assay showed a significant difference, with a low *Prevotella* load (Fig. 6C) and high *Bifidobacterium* load (Fig. 6D) in the fermented food-consuming groups when compared to the control Group-A, without any change in *Bacteroides* load (Suppl. Fig. S6). Moreover, the main archaeon, *Methanobrevibacter smithii*, did not differ between the two clusters of ecological states and did not also

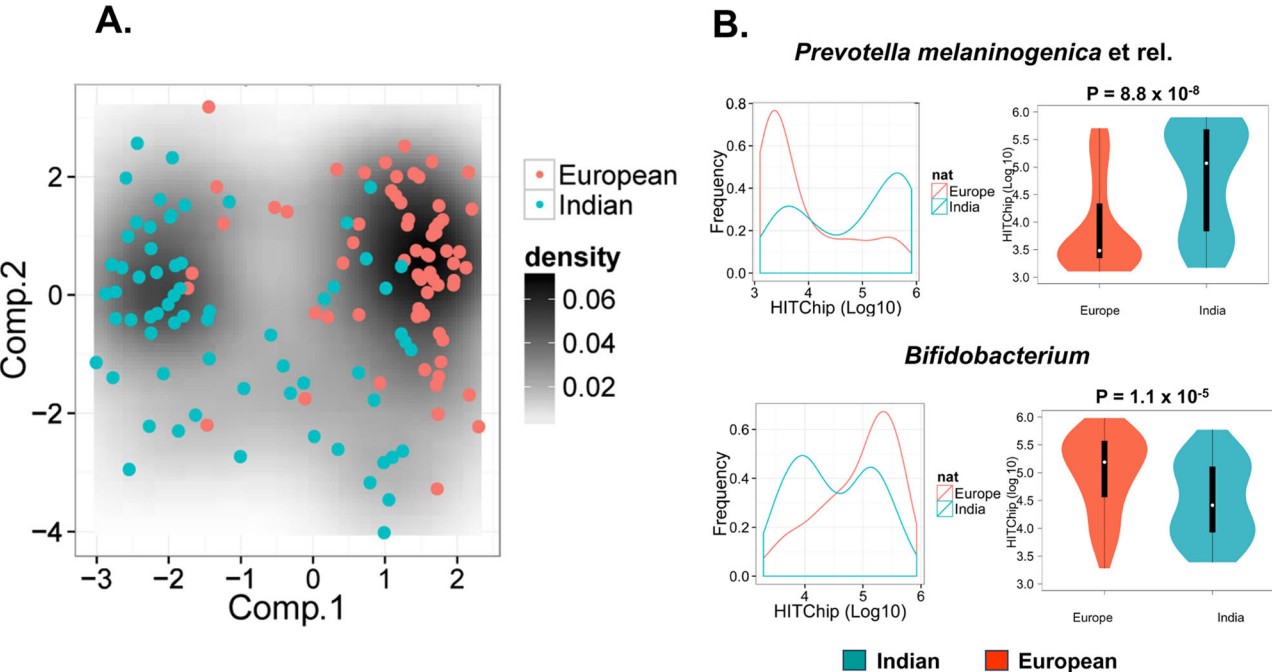

**Fig. 3 | Comparison of Indian gut microbiota with European. A** The PCA for Indian versus European gender, age, and BMI-matched samples shows the two modes associated with *Prevotella* status across the populations. The plot generated with the most prevalent 67 taxa for 76 subjects matched between European and Indian at baseline (summer), detected in >20% of the samples with HITChip probe signal above the detection threshold (> 2.8 log10). **B** The overall relative abundance histogram (density plots) and the violin plots of probe signals visualise the difference in the relative abundance distributions of *Prevotella melaninogenica* et rel. and

*Bifidobacterium* between European and Indian samples matched for sex, BMI, and age. The violin and whisker plots display the median (middle point), and the box ranges from 25 to 75 percentile with a whisker of lower than 1.5 IQR. The statistical significance of the difference between the biological replicates of Indian ($n = 76$) and European ($n = 76$) samples was analysed by a two-sided Wilcoxon test, and the $p$-values are indicated in the figure panel. The colour key for Europeans is orange-red and Indian in blue-green in all the figure panels. Source data are provided as a Source Data file.

differ in the diet groups (Suppl. Fig. S6). Our results supported that the frequent fermented food consumption in the long-term showed a difference in the gut bacterial compositions/ecological states, with the structure towards the *Bifidobacterium/Ruminococcus*-driven Cluster-B/R.

### No major change in the ingested bacterial load
The analysis on the fate of ingested bacteria by taxa-specific qPCR assay showed that the dominant bacteria present in the fermented foods (*Bacillus* of $10^{10}$-$10^{11}$ CFU g$^{-1}$ with $10.82 \pm 0.3$, mean ± s.d. log10 gene copies in fermented soybean *Hawaijar* and lactic acid bacteria of $10^{8}$-$10^{10}$ CFU g$^{-1}$ with $9.45 \pm 1.0$, mean ± s.d. log10 gene copies in fermented milk *Dahi*) represented a small proportion (*Bacillus* with $10^{4}$-$10^{9}$ CFU g$^{-1}$ with $5.93 \pm 0.9$, mean ± s.d. log10 gene copies and lactic acid bacteria with $10^{6}$-$10^{11}$ CFU g$^{-1}$ with $8.79 \pm 1.3$, mean ± s.d. log10 gene copies) in the gut bacterial load of $10^{10}$-$10^{12}$ CFU g$^{-1}$, $10.68 \pm 0.5$, mean ± s.d. log10 gene copies. Moreover, a similar bacterial load was already present in the gut bacteria of the control Group-A, which did not consume *Hawaijar* and *Dahi*. Our study did not observe any significant change in the lactic acid bacterial load in gut bacteria of *Dahi* consumers (Group-B and D) and seasonal differences in the *Bacillus* load in all diet groups. However, we observed a higher *Bacillus* load in the frequent *Hawaijar* consumer Group-C (Suppl. Fig. S7). Culture-independent analysis by Illumina-MiSeq amplicon sequencing of 16S rRNA gene resulted in *Bacillus* and lactic acid bacteria as the predominant bacteria in *Hawaijar* and *Dahi* respectively. The variability in the bacterial composition of *Hawaijar* and *Dahi* samples, five each, analysed by Illumina-Miseq amplicon sequencing was limited, showing occasionally other taxa being dominant in addition to *Bacillus* and lactic acid bacteria as the predominant bacteria in *Hawaijar* and *Dahi* respectively (Supplementary Tables S4 and S5).

### Limited changes in faecal bacteria and metabolites by fermented foods
PCoA visualization of sample similarities based on gut microbiota composition by Bray-Curtis dissimilarity indicated that the seasonal changes in gut bacterial composition (PERMANOVA: $F = 2.993$, $p = 0.0002$) were much higher than the impact of the diet groups (PERMANOVA: $F = 1.988$, $p = 0.0051$) in the Indian study population (Suppl. Fig. S8).

The diet groups (A, B, C, D) did not differ significantly in terms of sex (Fisher exact test, $p = 0.08$) or age (Kruskal-Wallis test $p = 0.76$), but differed significantly with BMI (Kruskal-Wallis test $p = 0.007$). As a high variability in the dietary habits recorded in individual subjects and seasonal variation in gut microbiota, we corrected the confounding factors by a Linear mixed effect model. We controlled the season and subject effects (age, sex, BMI) to visualise the significant diet effects (Group A-D), and controlled the diet and subject effects to visualise the seasonal effects. For this purpose, we used the Tukey post hoc test of the linear mixed model with a False discovery rate. The diet effects after controlling the confounding factors, the bacterial taxa with significant diet association (False discovery rate FDR < 0.25 adjusted $p$-value, BH corrected) are visualised in Fig. 7A and Suppl. Table S6. Among the significant diet-associated taxa, *Catenibacterium mitsuokai* et rel. was in lower absolute abundance in the *Dahi*-consuming groups (Group-B and Group-D) without seasonal changes (Group-A: $7.5 \pm 0.08$, Group-B and Group-D are $6.9 \pm 0.09$, $6.1 \pm 0.07$ respectively, median ± s.e.m, Log10 of 16S rRNA gene copies g$^{-1}$ of wet faeces) ($p < 0.001$, two-sided Wilcoxon test). Except *Catenibacterium mitsuokai* et rel., the taxa significantly changed by diet (*Prevotella tannerae* et rel., *Parabacteroides distasonis* et rel., *Tannerella* et rel., and *Bacteroides splanchnicus* et rel.) also varied significantly over season (Fig. 7B, Suppl. Fig. S9). The Random Forest analysis predicted the key

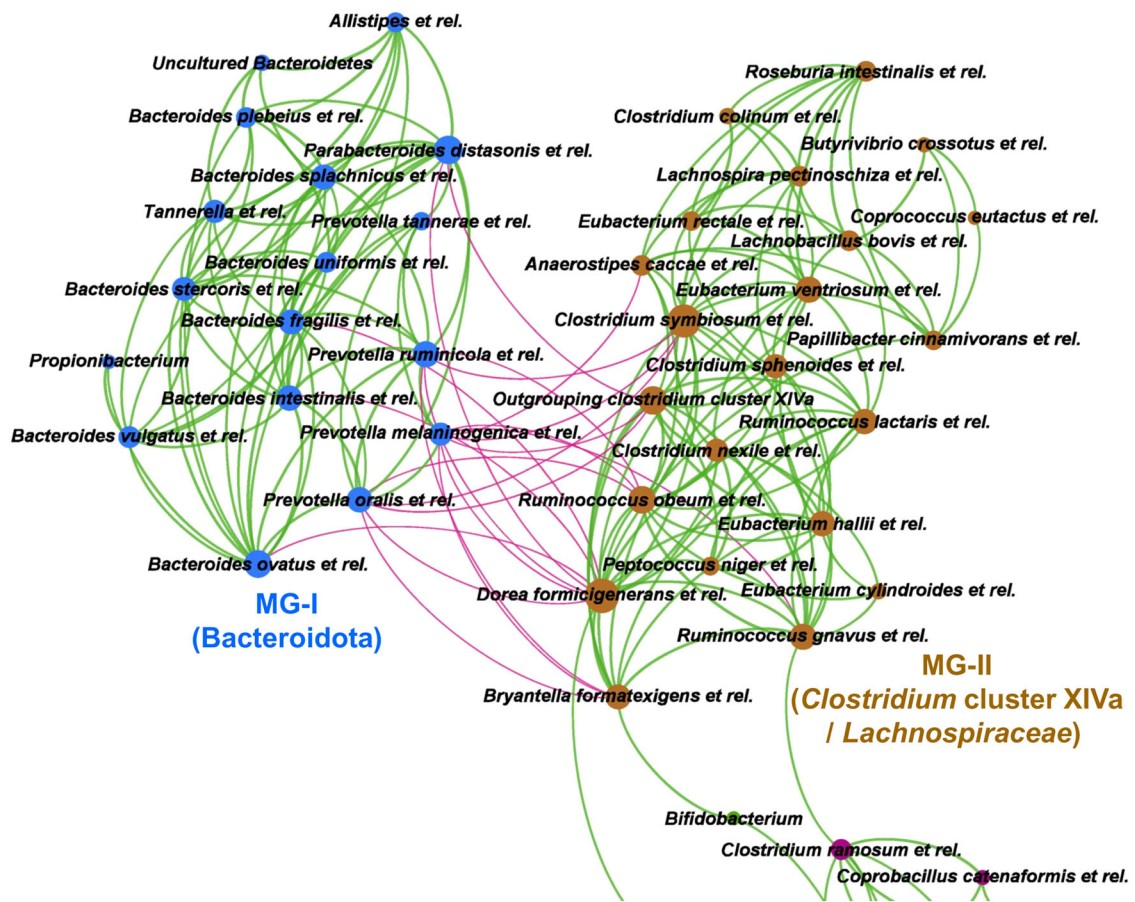

**Fig. 4 | Co-exclusion network between *Clostridium* cluster XIVa (*Lachnospiraceae*) and Bacteroidota visualised in the Indian study population.** A correlation network plot ($n = 214$) shows a negative association (Spearman correlation $<-0.4$, $p < 0.0001$) between two modularity groups of co-occurring taxa (Spearman correlation coefficient $>+0.4$, $p < 0.0001$), namely modularity group-I (MG-I) with members of Bacteroidota and modularity group-II (MG-II) with members of *Clostridium* cluster XIVa (*Lachnospiraceae*). The green line indicates a positive correlation, and the pinkish-red line indicates a negative correlation. The colour of the nodes indicates their modularity group assignment (blue colour for MG-I and brown colour for MG-II), and the size of the node indicates the number of edges adjacent to the node. Source data are provided as a Source Data file.

differentiating taxa (>25 MDA) as *Catenibacterium mitsuokai* et rel. in *Dahi* consumers, whereas *Coprococcus eutactus* et rel. in *Hawaijar* consumers (Suppl. Fig. S10). The *Coprococcus eutactus* et rel. showed significantly lower absolute abundance in the *Hawaijar* diet groups (Group-A: $9.18 \pm 0.5$, Group-B and Group-C are $8.51 \pm 0.5$, $8.12 \pm 0.5$ respectively, median $\pm$ s.e.m., Log10 of 16S rRNA gene copies g$^{-1}$ of wet faeces) ($p < 0.001$, two-sided Wilcoxon test). The significant diet-associated taxa (between Groups A, B, C and D) visualised in the above analysis were different from the significant BMI-associated taxa (Kendall's tau, Benjamini-Hochberg FDR-correction, p < 0.05) in the study population. The significant BMI-associated taxa were *Dorea formicigenerans* et rel. ($p = 0.017$), *Oscillospira guillermondii* et rel. ($p = 0.017$), *Ruminococcus gnavus* et rel. ($p = 0.017$), *Sporobacter termitidis* et rel. (0.017), *Clostridium symbiosum* et rel (0.029), *Eubacterium biforme* et rel. ($p = 0.0.29$), Uncultured Clostridiales II (0.029) and *Prevotella melaninogenica* et rel. ($p = 0.041$). Among the SCFA compared among the diet groups by HPLC analysis, we noticed a significantly higher production of acetate, propionate and butyrate ($p < 0.05$, two-sided Wilcoxon test) in the diet group-C and the higher mean of SCFA visualised in the diet group-D is not statistically significant (Suppl. Fig. S11). LC-HRMS faecal metabolites profile data did not show a difference between the diet groups. However, Random Forest analysis showed butyrate and phenyl hydrogen sulfate as key differentiating metabolites (MDA > 10) at a higher level in the *Hawaijar* consumers (Groups C and B), whereas glutarate at a higher level in

*Dahi* consumers (Groups B and D). Regarding the consumption of other fermented foods (apart from the two considered in the study), the study population consumed *Soibum* & *Soidon* (fermented bamboo shoots) and *Hentak* (fermented fish paste) with very little frequency, which were cooked before consumption. The 48-hour diet recall questionnaire data showed less consumption frequency of *Soibum* (24 out of 214), *Soidon* (3 out of 214), and *Hentak* (3 out of 214) in the study population. Confounder analysis did not relate the difference visualised in the diet groups with the consumption of *Soibum*, *Soidon* or *Hentak*.

## High changes in faecal bacteria during summer in fermented milk consumers

To understand the seasonal impact on faecal bacterial composition, we analysed the similarity changes of the paired samples over three seasons (Pearson correlation coefficient) and did not find any significant change ($p > 0.05$, ANOVA) in all the diet groups over three seasons (Suppl. Fig. S12). After controlling the diet and subject effects, the bacterial taxa showed significant seasonal changes (FDR < 0.25, Tukey post hoc test of the linear mixed model, BH corrected) to winter and summer with respect to the autumn season are shown in Fig. 7B, Suppl. Fig. S13. Most of the seasonal changes were in the Bacteroidota phylum, significantly changing to the lower relative abundance during summer (Suppl. Table S7). On the contrary, the lactic acid bacteria (*Lactobacillus*, *Streptococcus*, *Weissella*, *Aerococcus* and *Granulicatella*)

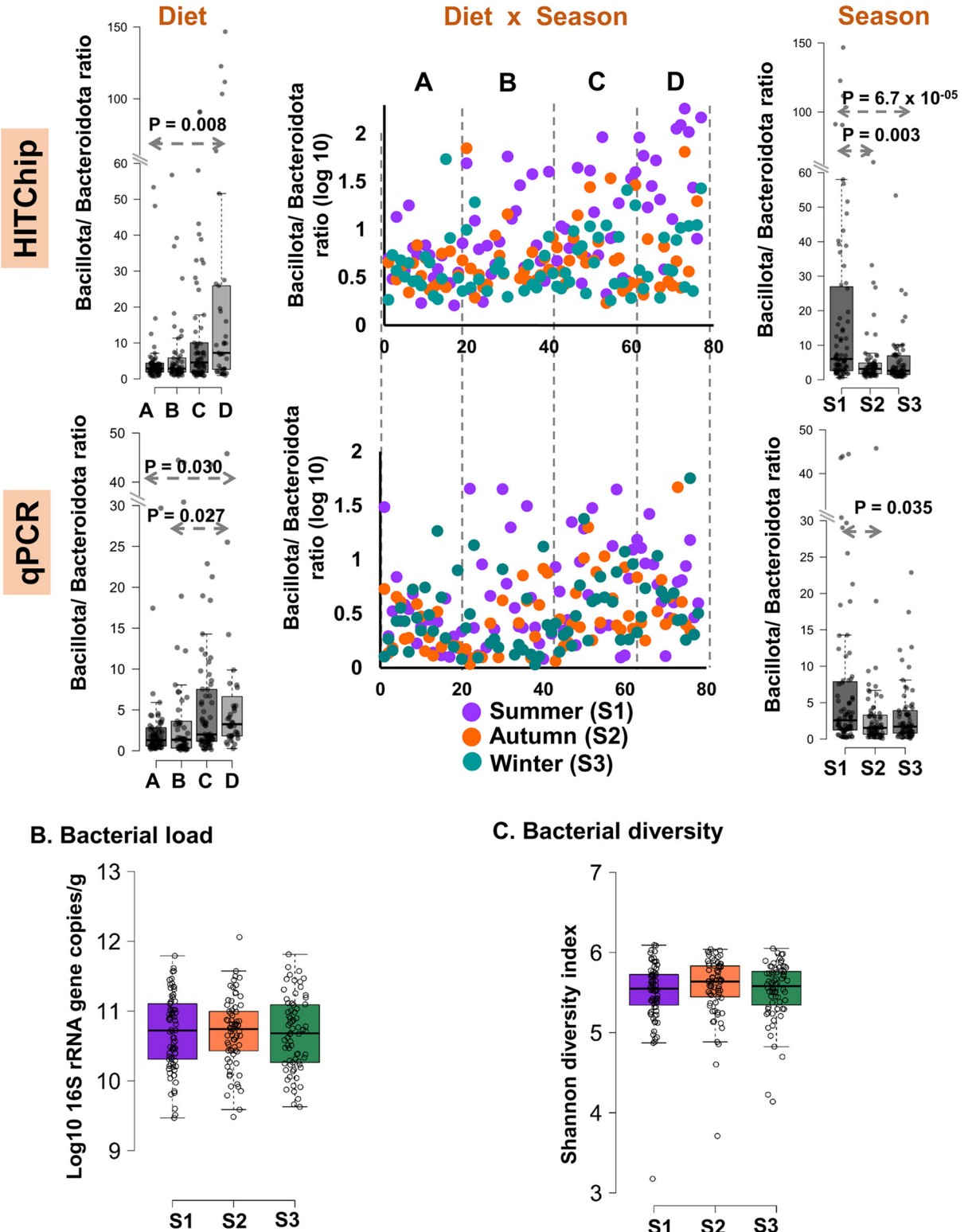

**A. Bacillota/ Bacteroidota ratio**

**B. Bacterial load**

**C. Bacterial diversity**

were in higher relative abundance during summer. *Subdoligranulum variabile* et rel. showed a significant seasonal fluctuation in control Group-A ($p = 0.012$, Wilcoxon, BH corrected) while stable in the fermented food consuming groups. Since the major discriminating taxa between the diet groups showed a seasonal fluctuation, we used Principal Response Curve (PRC) analysis to elucidate the overall gut bacterial compositional changes due to both diet and seasonal effects

together, and visualised the changes in the diet groups at three-time points by keeping the control Group-A as a baseline (Fig. 7C). PRC clarified that the diet impact was more ($p = 0.002$, Monte Carlo permutation test) during summer, particularly in the fermented milk consumers (Group-B and D). The changes due to the diets and seasons are explained mainly by the lower relative abundance of Gram-negative bacterial taxa of Bacteroidota and Pseudomonadota.

**Fig. 5 | Seasonal fluctuation of Bacillota/Bacteroidota ratio in the study groups without any change in the total bacterial load and bacterial diversity over the three seasons, analysed by HITChip and qPCR assays. A** The box and whisker plots with individual data point distribution show the changes in the Bacillota/Bacteroidota ratio between the four diet groups (A–D) and three seasons (S1, S2, S3), analysed by two different methods (HITChip and qPCR assays). The diet groups are Group-A: not consumed *Dahi* and *Hawaijar* (n = 58), Group-B: consumed *Dahi* and *Hawaijar* (n = 54), Group-C: consumed *Hawaijar*, not *Dahi* (n = 65), and Group-D: consumed *Dahi*, not *Hawaijar* (n = 37), and three seasons are S1- Summer (n = 76), S2- Autumn (n = 70), S3-Winter (n = 68). The box and whisker plots display median (middle line), box ranges from 25 to 75 percentile with Tukey whisker lower than 1.5 IQR. The scatter plot shows the change in the Bacillota/Bacteroidota ratio due to both the seasons and diet effects together (Diet × Season). The Bacillota/ Bacteroidota ratio is log10 ($x_i$+1) transformed for effective visualisation in the scatter plot. The significant changes between the diet groups and seasons were calculated by a two-sided Wilcoxon test, and Bonferroni corrected and indicated the p values in the figure panels. **B, C** The Box and whisker plots show no seasonal changes in the total bacterial load (**B**) and bacterial diversity (**C**) in the Indian study population. The total bacterial load (Total bacteria-specific qPCR assay) and the bacterial diversity (Shannon diversity index calculated from the HITChip probe profile) show no change between the three-time intervals. The box and whisker plots display median (middle line), box ranges from 25 to 75 percentile with Tukey whisker lower than 1.5 IQR, and no statistical significance between the seasons (ANOVA). The colour keys of the seasons are purple for summer (S1), orange-red for autumn (S2), and green for winter (S3). The Source data are provided as a Source Data file.

## Fermented foods and seasons impact the gut bacterial bimodality

We could visualise a different bimodal distribution (bimodality) in the gut bacterial taxa with significant diet and/or seasonal effects. Among the taxa with a significant diet effect, *Catenibacterium mitsuokai* et rel., *Prevotella tannerae* et rel., *Parabacteroides distasonis* et rel., *Tannerella* et rel., and *Bacteroides splanchnicus* et rel. showed a different bimodal distribution in the fermented food consuming groups (Group-B, C, and D) in comparison to the control diet Group-A, the changes were towards the lower relative abundance mode in the fermented food diet groups (Fig. 8, Suppl. Fig. S14). Interestingly, the seasonal bimodality differences in *Catenibacterium mitsuokai* et rel. visualised with a higher relative abundance mode during summer. The taxa of Bacteroidota (*Prevotella, Tannerella, Alistipes,* and *Bacteroides*), with significant seasonal effect, showed a change in bimodality with a distinct lower abundance mode during summer. However, the Bacteroidota were remarkably stable over three seasons in the control diet Group-A. In addition, *Bifidobacterium* showed a trend towards the lower relative abundance mode of bimodality during winter (Suppl. Fig. S15). The other taxa with a significant increase during summer showed an unimodal distribution. The analysis of inter-individual stability of taxa over seasons indicated that the bimodal taxa were generally more stable within individuals than the unimodal taxa. Overall results suggested that frequent fermented foods consumption in the long-term impacts the seasonal changes in the gut microbiota composition. The network of strong negative correlations visualised between the members of *Clostridium* cluster XIVa (*Lachnospiraceae*) and Bacteroidota in control Group-A, limited in the fermented food consuming groups, particularly in fermented milk consumers (Groups-B and D) (Suppl. Fig. S16). The above observations suggest that the co-exclusion network visualised here is a possible driver of gut bacterial stability. Frequent consumption of fermented foods may be one factor that could affect this interaction network, leading to seasonal changes in the ecological states of bimodality in the isolated rural Indian study population.

## Discussion

Though the gut microbiota of children (<3 years) and elders (>65 years) are highly variable, it is believed that the gut microbiota of healthy adulthood is relatively stable over time[33]. We observed a large degree of seasonal variation in gut microbiota (particularly Bacteroidota composition) in the Meitei community in Manipur, India, an isolated population of self-declared long-term fermented food consumers who have limited contact with other populations in India. However, the control group (Group-A), which did not consume fermented milk and soybean products over a minimum period of 10 years, did not show such seasonal variation in Bacteroidota composition. The seasonal change in gut microbiota observed in our study is similar to the seasonal cyclic variation observed by Smits et al. (2018)[12] in Hadza hunter-gatherers of Tanzania. The impact of seasonal weather changes on gut microbiota may be more visible in the traditional population living in an unconditioned household environment

(without proper heating, cooling or insulation facilities), which is very much exposed to outside temperature changes, similar to the present study population households. We observed a decline in the relative abundance of Bacteroidota during the hot-humid summer compared to the cold-dry winter (Suppl. Fig. S17) in the Indian study population. The seasonal dietary differences are explained as the potential drivers of this temporal instability in human gut microbiota[12]. Also, several researchers linked the difference in Bacteroidota composition with the changes in energy expenditure[34], fat metabolism[35] and physiological disorders[7]. Our finding of seasonal changes in gut microbiota and metabolites composition seems to be important concerning the seasonal occurrence of some non-infectious diseases. Particularly, seasonal changes in mental health like depression and anxiety and modulation of related gut microbiota by fermented foods are important areas for future research[36].

A review by Dimidi et al.[37] shows limited evidence of the impact of fermented foods and probiotics on gut microbiota composition in healthy adults[25,26]. Recently, a few studies observed differences in the gut bacterial composition between milk and fermented milk consumption[38] and decreased gut bacteria diversity similar to our results due to fermented goat milk[39]. Although a healthy gut microbiota is difficult to describe, ecological stability, i.e. elasticity to diet or environmental stress extremes[40] and high bacterial diversity, is considered important. Many studies correlated the changes in gut bacterial diversity and Bacillota/Bacteroidota proportion with health[7]. In our study, long-term fermented food consumption showed seasonal instability in Bacteroidota and reduced gut bacterial diversity. Is this high seasonal instability a positive or negative trait? Though high bacterial diversity is an indicator of colon health, high diversity and Bacteroidota enrichment correlated with several diseases[41], including inflammatory bowel diseases[7]. The seasonal instability of gut bacteria due to long-term frequent fermented foods consumption may allow the gut more colonization permissive and weaken its resilience nature. However, it also cautions against frequent consumption of fermented foods in long-term loss of gut bacterial diversity and elasticity and may lead to a disturbed condition.

The key differentiating metabolites between the bimodality states of gut microbiota (Cluster-P and Cluster-B/R) are mostly the derivatives of long-chain fatty acids and certainly butyrate, suggesting a shift in lipid metabolism. Similar to our study, Taylor et al.[42] recently showed the enrichment of conjugated linoleic acid (CLA) in the faecal metabolites of fermented food consumers. Moreover, the bacteria abundantly present in the *Bifidobacterium/Ruminococcus*-driven Cluster–B/R are well known for conjugated fatty acid production[43]. Also, higher Ciprofibrate (a lipid-lowering agent[44]) and Linagliptin (hypoglycemic and neuroprotective agent[45]) production support the difference in lipid metabolism in Cluster-B/R and related health benefits. High butyrate production and enrichment of *Bifidobacterium* and *Lachnospiraceae*[12] also support the health benefits of Cluster-B/R. Higher acetate production during fermented food consumption possibly helps to prevent pathogen establishment[46], but more acetate

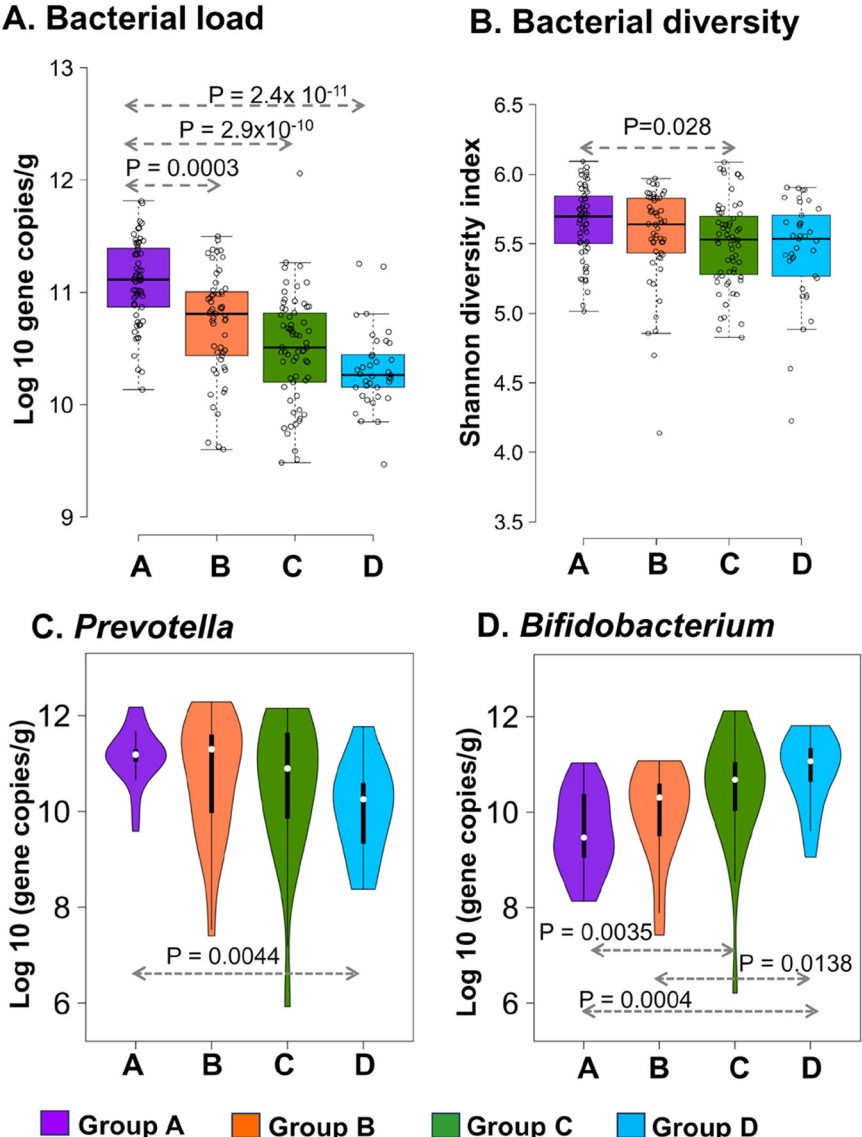

**Fig. 6 | Effect on the overall gut bacterial load, Shannon diversity, *Prevotella* and *Bifidobacterium* load during long-term fermented foods consumption.**
**A**, **B** The box and whisker plots show a lower bacterial load (**A**) in the fermented food consuming groups (Group-A: not consumed *Dahi* and *Hawaijar* (*n* = 58), Group-B: consumed *Dahi* and *Hawaijar* (*n* = 54), Group-C: consumed *Hawaijar*, not *Dahi* (*n* = 65), and Group-D: consumed *Dahi*, not *Hawaijar* (*n* = 37), quantified by bacteria-specific qPCR assay and expressed as 16S rRNA gene copies/g of wet faeces, and lower Shannon diversity (**B**) calculated from the HITChip probe profile in the fermented food consuming groups. The box and whisker plots display the median (middle line), and the box ranges from 25 to 75 percentile with Tukey whisker (less than 1.5 IQR). **C, D** Violin and whisker plots show a change in the

*Prevotella* load (Group A: *n* = 20, Group B: *n* = 20, Group C: *n* = 20, Group D: *n* = 14) and *Bifidobacterium* load (Group A: *n* = 20, Group B: *n* = 16, Group C: *n* = 19, Group D: *n* = 13) in the fermented food-consuming groups, quantified by taxa-specific qPCR assay. The bacterial load is expressed as 16S rRNA gene copies/g of wet faeces. The violin and whisker plots display the median (middle point), and the box ranges from 25 to 75 percentile with a whisker lower than 1.5 IQR. The statistical significance of variation between the study groups was calculated by a two-sided Wilcoxon test, Bonferroni corrected and indicated as *p*-value in the figure panels. The colour keys of the diet groups are Group-A (purple), Group-B (orange-red), Group-C (green), and Group-D (sky blue). The source data are provided as a Source Data file.

production may increase the lipid droplet formation in the intestinal epithelium and liver[47]. Moreover, the taxa reduced during fermented milk consumption (*Catenibacterium* and Bacteroidota) are linked with bad fat metabolism (positively correlated with LDL cholesterol level)[43]. Our observation of fatty acids as the key differentiating metabolites between Cluster-P and Cluster-B/R suggests a change in the overall lipid metabolism, which may change the blood fat composition health. The significantly high BMI in subjects of Cluster-B/R support the above observation. The bimodal distribution of *Catenibacterium* showed a change towards the high abundance mode during summer, while to the low abundance mode during frequent *Dahi* consumption. Similarly, *Bifidobacterium* showed a low abundance mode during winter,

while *Dahi* and *Hawaijar* consumption changed it towards a higher abundance mode. The above observations support the traditional beliefs of the body cooling effect by *Dahi* during summer and winter wellness by *Hawaijar*.

*Prevotella*, the most abundant gut bacteria of the Indian study population, shows the lowest genetic potential of producing carbohydrate-active enzymes (CAzymes) and SCFA among the taxa of phylum Bacteroidota[48]. Therefore, variation in *Prevotella* may result in a substantial difference in the capacity to digest complex carbohydrates in-between the two ecological clusters. High level of butyrate in *Bifidobacterium*/*Ruminococcus*-driven Cluster−B/R support gut health in preventing inflammation and

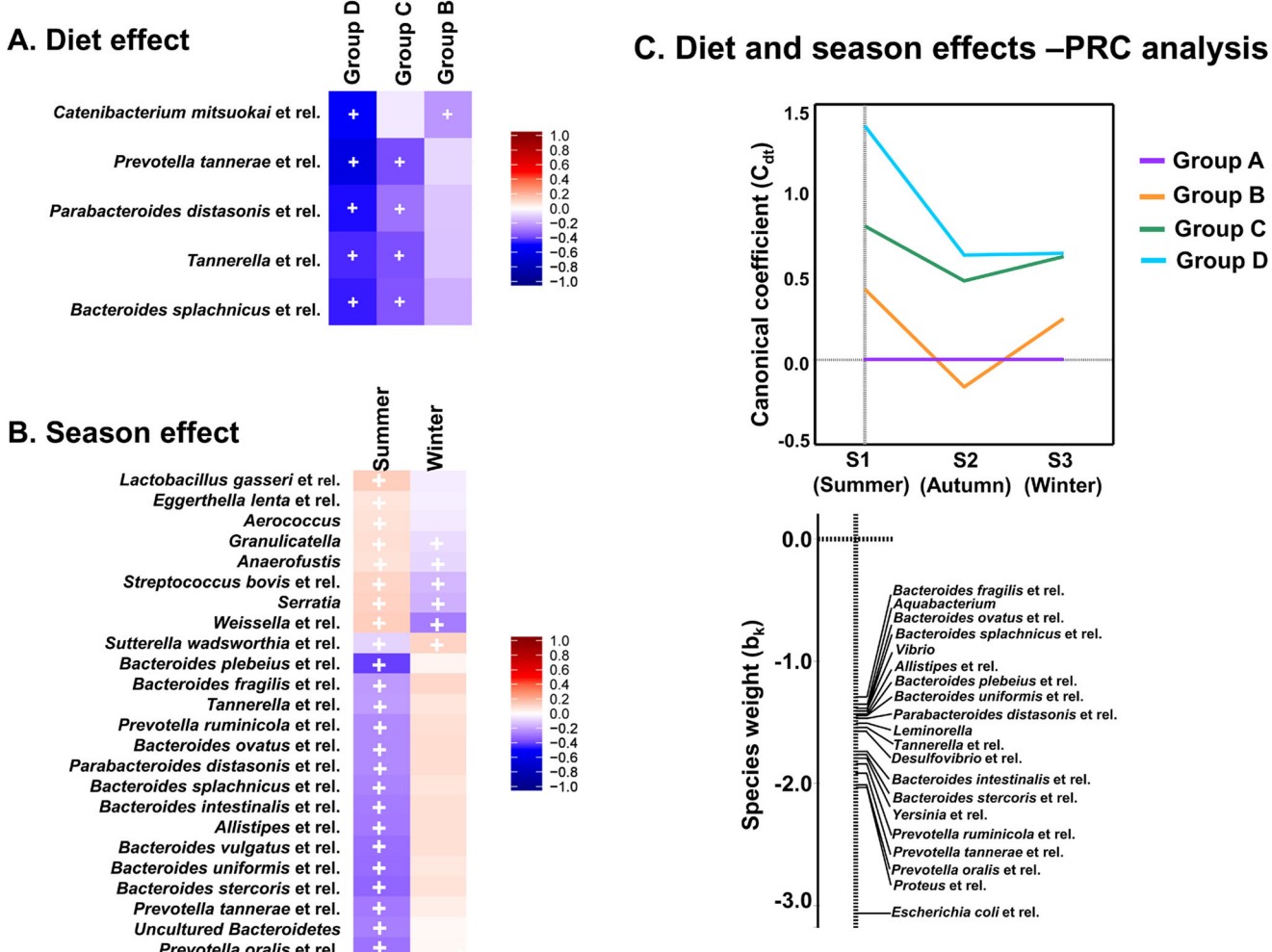

**Fig. 7 | Gut microbiota composition changes due to the diet (long-term consumption of fermented foods) and seasonal changes. A** The heat map shows the gut bacteria with significant diet effects (with respect to the control diet, Group-A) from the Tukey post hoc test of the linear mixed-effects model after controlling the season and subject effects. The coefficient shows how much the diet alters the taxa's relative abundance on average with respect to the control diet (Group-A). The diet groups are Group-A ($n = 58$): not consumed *Dahi* and *Hawaijar*, Group-B ($n = 54$): consumed *Dahi* and *Hawaijar*, Group-C ($n = 65$): consumed *Hawaijar*, not *Dahi*, and Group-D ($n = 37$): consumed *Dahi*, not *Hawaijar*, and three seasons are Summer ($n = 76$), Autumn ($n = 70$), and Winter ($n = 68$). **B** The heat map shows the gut bacteria with significant seasonal effects (with respect to autumn) after controlling diet and subject effects. The significant associations of taxa were calculated by paired two-sided Wilcoxon test, matched for age, BMI and gender; Benjamini-Hochberg corrected $p$-values, FDR < 0.25, absolute fold-change >2 of HITChip signals and highlighted with '+'. **C** The principal response curve (PRC) analysis shows the overall composition change in gut microbiota due to both seasonal and diet effects together. The fluctuating Canonical coefficient (Cdt) as taxa level changes ($p = 0.002$, two-sided Monte Carlo permutation test, $n = 499$) during three-time points by keeping the control group (Group-A) as a baseline is shown here. Source data are provided as a Source Data file. Colour-coded by the group assignment. Negative values of species weight indicate a low relative abundance of discriminating taxa in the fermented food consuming groups.

immunosenescence. *Alistipes* and *Parabacteroides* produce butyrate via amino acid fermentation, whereas *Lachnospiraceae* produced it via butyryl CoA:acetate CoA-transferase pathway[49]. Therefore, the change in the ecological state towards Cluster-B/R may change the versatility and plasticity in polysaccharide degradation and butyrate production. Several studies linked *Prevotella* enrichment in the gut bacteria with chronic inflammation[50,51], and the countries reported with *Prevotella*-dominated gut bacteria are often malnourished. Moreover, the negative correlation of *Prevotella* enrichment with longevity predicted across genotypically non-related centenarians[52] sustains the hypothesis of its negative association with health. We question whether *Prevotella* enrichment in the traditional population indicates malnutrition or gut dysbiosis, or chronic infection? *Prevotella* enrichment in gut microbiota is reported in the countries of the tropical region. Is there any role of hot-humid climate in shaping gut microbiota composition? Is there chronic *Prevotella* infection in tropical countries? Could fermented

foods improve this condition by changing the ecological state towards *Bifidobacterium*/*Ruminococcus*-driven Cluster–B/R?

We report bimodality in gut microbiota structure and a difference in the modes during seasons in a rural Indian population. The co-exclusion network visualised between *Clostridium* cluster XIVa and Bacteroidota is possibly the driving force for this bimodality and seasonal variation in gut microbiota. Cooperative warfare[53] is the common way of maintaining this co-exclusion network in microbial ecology. Our results indicated that fermented food consumption might affect the co-exclusion network between Bacteroidota and *Clostridium* cluster XIVa (Suppl. Fig. S16). Fermented foods that support Bacillota growth might have favoured *Clostridium* cluster XIVa. The upward meta-transcript changes in *Clostridium* cluster XIVa during fermented milk consumption noticed by McNulty et al.[26] supported our observation. Frequent consumption of fermented foods in the long-term, particularly fermented milk, might have strengthened the *Clostridium* cluster XIVa modularity

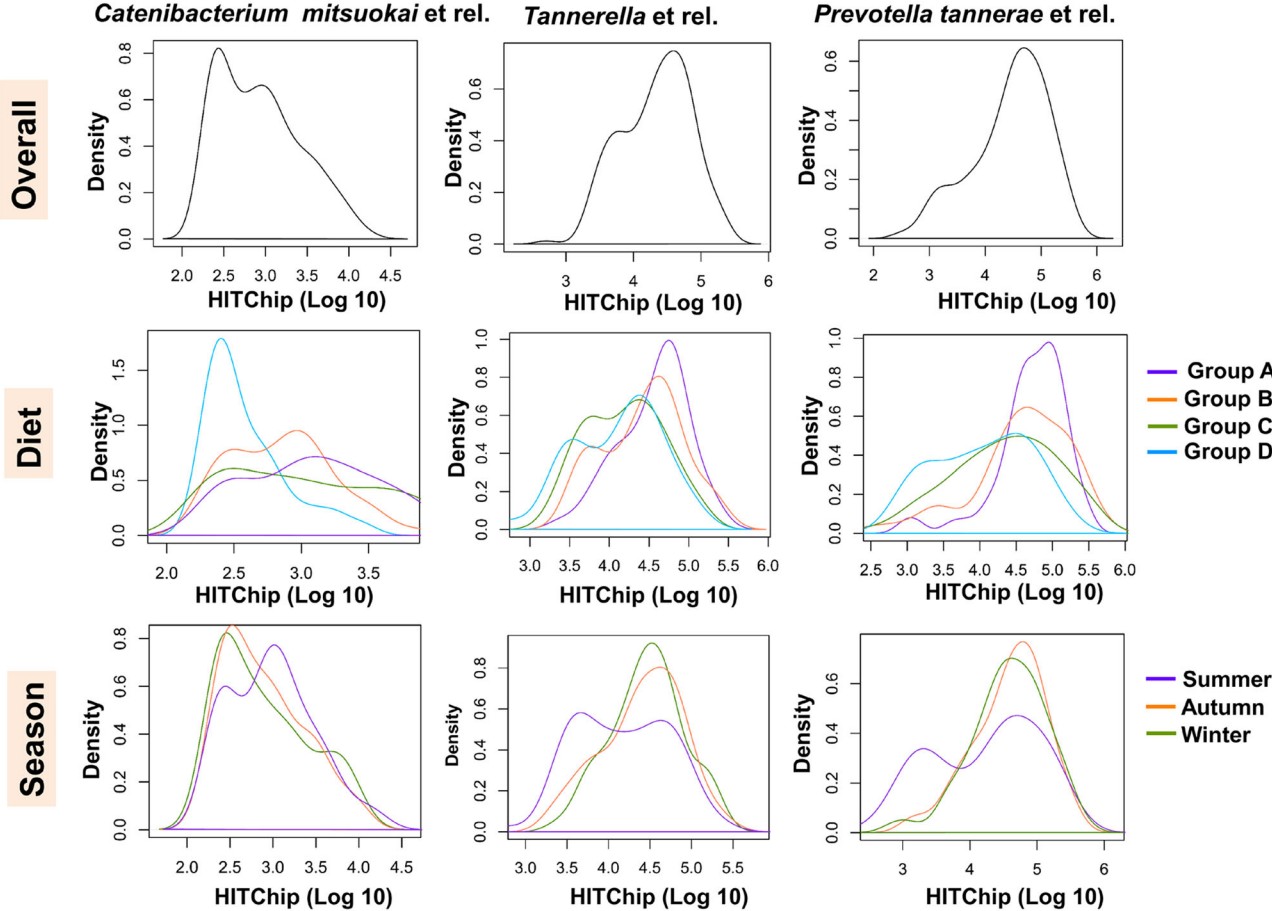

**Fig. 8 | Changes in the bimodal distribution of *Catenibacterium mitsuokai* et rel., *Tannerella* et rel., and *Prevotella tannerae* et rel., with significant diet association between four study groups (Diet) and three seasons (Season).** The density plots show a shift in the bimodal distribution due to diet and seasonal effects. The diet groups are Group-A: not consumed *Dahi* and *Hawaijar* (*n* = 58), Group-B: consumed *Dahi* and *Hawaijar* (*n* = 54), Group-C: consumed *Hawaijar*, not *Dahi* (*n* = 65), and Group-D: consumed *Dahi*, not *Hawaijar* (*n* = 37), and three seasons are Summer (*n* = 76), Autumn (*n* = 70), and Winter (*n* = 68). The statistical significance of the taxa with diet associations was calculated by Tukey post hoc test of the linear mixed-effect model with false discovery rate FDR < 0.25, Benjamini-Hochberg method adjusted *p*-value. Source data are provided as a Source Data file.

group (*Lachnospiraciae*), which may affect the co-exclusion network and led to the seasonal instability in Bacteroidota. The earlier reports of a significantly lower proportion of Bacteroidota during fermented milk consumption[54,55] indicate a similar impact visualised in our study. We speculate that the trend of increased *Bifidobacterium* by fermented soybean[56] and fermented milk[57], increased butyrate production by taxa within the *Clostridium* cluster XIVa[58] and increased gene expression of *Clostridium* cluster XIVa[26] by fermented milk with probiotics support a change in gut microbiota composition towards *Bifidobacterium*/*Ruminococcus*-driven Cluster–B/R. We observed that nearly one-fourth of the studied individuals had shifted their gut bacterial ecological state over six months. Similar temporal instability between two gut bacterial clusters was also observed in an earlier study[59]. A trade-off between *Prevotella* and *Bifidobacterium* enrichment was also evidenced in our study. Moreover, a higher presence of *Bifidobacterium* and lower presence of *Prevotella* and *Catenibacterium* in the European population compared to this study looks like a possible change in the gut microbiota structure towards *Bifidobacterium*/*Ruminococcus*-driven Cluster–B/R due to frequent consumption of fermented foods. The higher relative abundance of *Bifidobacterium* during the infant stage, more *Clostridium* cluster XIVa during the adult stage, more Bacteroidota, and reduced *Clostridium* cluster XIVa during the elder stage[60] indicated that the co-

exclusion network observed here may play a role in the developmental biology.

The bimodal distribution (Cluster-P with *Prevotella* enrichment; Cluster-B/R with *Bifidobacterium*/*Ruminococcus* enrichment) was also visualised when the gut microbiota across geography was put together[12]. This difference in bimodality is explained as industrialised vs traditional population or developed vs underdeveloped population. The gut microbiota of the traditional population of Papua New Guinea (PNG)[61], Hadza hunters[12], and Amerindian[62] enriched with *Prevotella* and depleted with *Bifidobacterium* and *Lachnospiraceae*[61]. A similar difference was also visualised between the rural and urban populations in Russia[63] and India[64]. Interestingly, the countries of *Prevotella* enrichment placed with low UNDP human development index (HDI) and low WHO healthy life expectancy. All metagenomics studies show the disappearance of gut bacterial diversity in industrialised countries compared with traditional societies. Loss of gut bacterial diversity was also found to occur in individuals who migrated from traditional food culture to Western culture[65]. The western-associated *Bacteroides* started to displace the non-western-associated *Prevotella*[18]. Continuous displacement of *Prevotella* to exposure to a Westernised diet over decades of US residence was also noticed[66]. Fermented foods are a rich source of hydrolytic enzymes, which can simplify the complex diet. The long-term food habit of fermented food and processed foods might have changed the gut microbiota composition of present status

in developed countries. In such cases, it is important to understand the ecological and functional consequences of changes in the gut microbiota composition towards Cluster-B/R in developed countries.

In conclusion, we observed a seasonal instability of Bacteroidota during the frequent long-term consumption of fermented foods in a geographically isolated Indian population and noticed a seasonal change in the gut microbiota's ecological state and metabolism. Our study also cautions the ecological and functional consequences of the continuous displacement of *Prevotella* and loss of gut microbiota diversity due to frequent long-term consumption of fermented foods. We propose the interaction network between Bacteroidota and *Clostridium* cluster-XIVa visualised in our study could be a possible driver of seasonal instability in gut microbiota, particularly in long-term fermented food consumers. The relatively small size of samples analysed here limits the strength of our study. However, the seasonal instability in the human gut microbiota due to long-term fermented food consumption observed in our study can be taken forward for future studies with large-scale subject recruitment and sample analysis. Further understanding of the mechanism of seasonal instability in gut microbiota by fermented foods will allow strategies to modulate the gut microbiota composition towards or sustain a healthy status.

## Methods

### Ethics approval

The study was conducted according to all relevant ethical regulations required to research human participants as per the guidelines of the Indian Council of Medical Research (ICMR) (https://ethics.ncdirindia.org/icmr_ethical_guidelines.aspx). The study received ethical approval from the Institutional Ethical Committee (IEC) of the Institute of Bioresources and Sustainable Development, Imphal, India (approval number IBSD/IEC/2018/003), and the Department of Biotechnology, Government of India approved the study protocol. Written informed consent was obtained from all the participants, for the children from the children's parents or legal guardians. Samples were collected on a voluntary basis, and no compensation was provided for the participants.

### Inclusion & ethics statement

The study's first author is from the local research institute in India (IBSD), and he was involved throughout the research process. The research was undertaken with the higher standards of following local regulations. All collaborators of this study fulfilled the criteria for authorship, and roles and responsibilities were agreed upon ahead. This work's findings are locally relevant and have been determined with local partners. The main part of this research is the capacity-building of a local researcher (Jeyaram, K.) for the advanced molecular techniques at Wageningen University (The Netherlands). This research was not severely restricted or prohibited and did not result in discrimination or personal risk to participants. A Material Transfer Agreement (MTA) was signed between IBSD (India), and the Laboratory of Microbiology, Wageningen University (The Netherlands) to share benefits and describe roles and responsibilities. The Department of Biotechnology (DBT), Government of India, approved the material transfer agreement. The overall study was much safer; only healthy subjects' faecal samples were analysed, and all precautions and safety measures were undertaken for sample handling without isolating the microbes. Local and regional research relevant to our study, related to Indian fermented foods and Indian gut microbiota research, was considered in the citations.

### Study subjects and sampling

A survey was conducted over thousands of people in a genetically and culturally homogenous population (Meitei community, Mongoloid race) across 25 sq. km areas of Imphal valley regarding the frequency and quantity of fermented foods consumption. The target population was endogamous and had similar dietary and lifestyle habits. From the survey, four study groups (Group-A, Group-B, Group-C and Group-D) were identified based on the self-reported long-term dietary habit of consuming fermented foods, namely fermented soybean (*Hawaijar*[31] containing *Bacillus* load of $10^{10}$-$10^{11}$ CFU g$^{-1}$) and fermented milk (*Dahi*[30] contain the lactic acid bacterial load of $10^8$-$10^{10}$ CFU ml$^{-1}$). Group-A, never consumed the above two fermented foods at least for the last ten years, was considered as control; Group-B consumed both the fermented foods (*Dahi* and *Hawaijar*), Group-C consumed *Hawaijar*, not *Dahi*, and Group-D consumed *Dahi*, not consumed *Hawaijar*; with a consumption frequency of 3-7 times week$^{-1}$ @ 50-100 ml time$^{-1}$ for *Dahi* and @10-20 g time$^{-1}$ for *Hawaijar*. Though the frequency of fermented foods consumption varied over seasons, with a higher frequency (even daily) of fermented milk *Dahi* consumption during summer and fermented soybean *Hawaijar* during winter, the consumption frequency of a minimum of three times per week reported during the other seasons. The selection was made with the eligibility criteria of good general health, normal bowel frequency, free from any gastrointestinal diseases and other diseases, not taken antibiotic within six months before sampling. With the above eligibility criteria, about 20 subjects in each categorised study groups were targeted (Post-hoc power analysis with a power of >0.85, https://homepage.univie.ac.at/robin.ristl/samplesize.php?test=ttest). The subjects of four diet groups were randomly selected (double-blind randomization) from the list of subjects qualified with the selection criteria and used for the study. The study groups were balanced in age (< 14 years: 6, 15-24 years: 10, 25-34 years: 20, 35-44 years: 16, 45-54 years: 16, 55-64 years: 7, >65 years: 3), sex (self-reported biological sex, Male: 40, Female: 38), body mass index (BMI), nature of birth, dietary and lifestyle habits (Suppl. Table S1; Suppl. Results). Temporal sampling over three different seasons, hot-humid summer (S1), autumn (S2) and cold-dry winter (S3), were collected from the study population (Suppl. Fig. S17). The faecal samples were collected at home. The IHMS Standard procedure IHMS SOP 004 (https://human-microbiome.org) was followed for the sample collection by using Sterile Clinicol (Himedia). The self-collected faecal samples were frozen immediately by using a prechilled gel pack or dry ice in a cooler box and transported to the laboratory within two hours of collection and stored at −80° C, and extracted the DNA within seven days of sample collection. The subjects who took antibiotics during the sampling were dropped from the study groups. A questionnaire was used to get the metadata of the study subjects, including body height, weight, sex, nature of birth, marital status, long-term dietary habits and lifestyle habits. Furthermore, 48-hour diet recalls, health status and medication, if any, during each sampling were also recorded. To compare the bimodal distribution of the gut microbiota observed in the Indian study population, a healthy European population (n = 76 subjects from across Europe) matched for sex, BMI, and age was randomly selected from the HITChip database[32].

### DNA extraction and HITChip analysis

The DNA from faecal samples was extracted by 0.1 mm zirconia beads beating and QIAmp DNA kit (Catalog #56304, Qiagen) as described by Salonen et al. (2010)[67]. The faecal microbial diversity and composition were analysed by using the Human Intestinal Tract chip (HITChip), a phylogenic microarray as described by Rajilic-Stojanovic et al. (2009)[6]. Briefly, the DNA was used for PCR amplification of bacterial 16S rRNA gene by using primers T7prom-Bact-27-for (5'-TGAATTGTAATA CGACTCACTATAGGGGTTTGATCCTGGCTCAG–3') and Uni-1492-rev (5'-CGGCTACCTTGTTACGAC-3'), subsequently in vitro transcription (RNAMAXX T7 Transcription kit, Stratagene, Catalog #200339) and Cy3 and Cy5 post-labelling (GE healthcare, RPN5661). The HITChip microarrays (Agilent Technologies, Palo Alto, CA., USA) were hybridised by Cy3/Cy5 labelled targets, scanned using a microarray scanner (Agilent), and the signal intensity data from microarray images were

extracted using Agilent feature extraction software, version 9.5 (http://www.agilent.com). Data normalisation, baseline correction and further analysis were performed in a custom-designed MySQL database management system (http://www.mysql.com/)[6]. Sum and Robust probabilistic averaging (RPA)[68] were used to aggregate the data to the higher phylogenetic levels from the probe-level signals. The HITChip probe signals were assigned to three different phylogenetic levels. Level 1: order-like 16S rRNA gene sequence groups and level 2: genus-like with sequence similarity of >90%. The same protocol of faecal DNA extraction and the HITChip pipeline was used for the European population gut microbiota profiling for the comparative analysis, allowing an accurate one-to-one comparison between the gut microbiota from both regions. The data was imported in the R/Bioconductor *TreeSummarizedExperiment* data container for downstream analysis with *mia* (v. 1.15.6; https://www.bioconductor.org/packages/devel/bioc/html/mia.html).

## Metabolic profiling by LC-HRMS analysis

The ice-cold extraction solvent (1:1 v/v acetonitrile and water) homogenised faecal sample was passed through a 0.2 μm syringe filter (PTFE, Merck) and subjected to LC-HRMS analysis[69]. The untargeted faecal metabolite profiling was performed by a Dionex Ultimate 3000 ultrahigh-performance liquid chromatography (UHPLC) coupled with Q-Exactive Orbitrap (Thermo Fisher Scientific) (C-CAMP, Bengaluru). The hydrophilic interaction liquid chromatography column (HILIC, 5μ, 150 mm x 4.6 mm, Phenomenex Luna) at 40 °C with a mobile phase of 5 mM ammonium acetate in water (phase A) and 5 mM ammonium acetate in water with acetonitrile in a ratio of 1:9 (phase B), with a flow rate of 0.4 mL/min, in a 45 min gradient of 100 % to 0% Phase B 10 was used[52]. Human Metabolome DataBase/Kyoto Encyclopedia of Genes and Genomes (HMDB/KEGG) was used for the data analysis. The identity of the compounds that significantly differ between the study groups was assigned by analysing the acquired MS/MS spectra using the spectral database mzCloud. The statistical significance in metabolite level was analysed by Wilcoxon test, BH corrected $p$-values. A hierarchically clustered heat map of LC-HRMS metabolite data (with >2 log fold difference) was visualised using R library "gplots" using custom R scripts.

## Cluster analysis

The samples of the Indian study population were hierarchically clustered (complete hierarchical clustering based on Pearson correlation) based on the HITChip probe signals (log 10 transformed oligo-level relative abundance profiles). The similarity in gut microbiota composition was visualised using a heatmap. Subsequently, the overall gut bacterial community composition was analysed by an unsupervised Principal Coordinate Analysis (PCoA; of Bray-Curtis dissimilarity) for the genus-level profiles using PAST v3.22[70]. Similarly, the faecal metabolites data generated by LC-HRMS analysis were quality filtered, normalised, log-transformed (log xi + 1) and visualised with principal component analysis (PCA). The significance in the bacterial community composition or metabolite profile difference between the bimodal groups, the diet groups and seasons was calculated by PERMANOVA test with 9999 permutations using Bray-Curtis dissimilarity using PAST, and indicated as Bonferroni corrected $p$-values. The relative abundances were calculated with total sum scaling (TSS) transformation; the data was log10-transformed and compared between the categorised groups. The statistical significance of all the comparisons of taxa or metabolite between the groups were assessed with Wilcoxon signed-rank tests and expressed as BH corrected $p$-values using R. The supervised redundancy analysis (RDA) was performed using Canoco software v4.52 (Wageningen University, The Netherlands) with Monte-Carlo permutation test ($n = 499$) for finding the significance in the gut bacteria association with metadata. A hierarchically clustered heat map was generated with complete hierarchical clustering of faecal metabolites that are significantly differed (~2 log2 fold change, $p < 0.05$, BH corrected) between the two clusters by R package-based scripts. For comparing Indian ($n = 76$, at baseline summer) and European ($n = 76$) populations, the subjects were matched for sex, BMI, and age. PCA was performed with the most prevalent gut bacterial taxa, detected in >20% of the samples with >2.8 log10 HITChip probe signal. Further, the taxa that are significantly different ($p < 0.05$, BH adjusted; >2 absolute fold-change) between European and Indian samples were visualised with density plots (histogram) and line plots (comparison between the matched samples) to show the distribution difference.

## Diversity analysis

The Shannon diversity index of the gut bacteria was calculated from the relative abundance normalised signal values of all HITChip probes. The statistical significance of the difference in bacterial diversity between the categorised groups (two clusters, four diet groups, three seasons, Indian and European) was calculated using the Wilcoxon test with Bonferroni correction.

## Bacterial load by qPCR assays

The taxa-specific qPCR assays were used to calculate the difference in the absolute bacterial load between the categorised groups and expressed as gene copies/g of wet faeces. The primer sequences, conditions and standard bacterial strains used for quantifying the total bacterial load, Bacillota, Bacteroidota, lactic acid bacteria, *Bacillus subtilis*, *Prevotella*, *Bifidobacterium*, *Bacteroides* and *Methanobrevibacter smithii* are listed in Suppl. Table S8. The qPCR assays were performed in Biorad CFX384 (Biorad, USA) and ABI 7500 instrument (Life Technologies, USA) using SYBR green chemistry. Standard graphs were constructed using known copies of the 16S rRNA gene in the plasmid clone (pGEM-T, Promega, A1360). An eight-point standard graph range of $10^1$–$10^8$ gene copies with an assay efficiency of more than 0.99 was used, and all the samples were run in triplicates. The absolute load of other bacterial taxa that were significantly different between study groups was calculated by multiplying the HITChip relative abundance data with the total bacterial load in the sample calculated by total bacteria-specific qPCR assay. The qPCR data on the absolute quantification of respective taxa were visualised as a boxplot using BoxPlotR (http://shiny.chemgrid.org/boxplotr/). The violin plots were generated using PAST 3.22 software to show the difference in the bimodal distribution of *Prevotella* and *Bifidobacterium* between the categorised groups. Twenty samples each of naturally fermented milk (*Dahi*) and naturally fermented soybean (*Hawaijar*) were collected from the local markets of Imphal, Manipur, India for quantification of lactic acid bacteria and *Bacillus* load, respectively. The DNA was extracted by enzymatic lysis principle; the food homogenate pellet was treated with lysozyme (50KU) and mutanolysin (25 U), proteinase K, and finally precipitated with isopropanol as described by Keisam et al. (2016)[71,] and taxa-specific qPCR assays were performed. The statistical significance of the difference between the study groups was calculated by a two-sided Wilcoxon test and expressed here as $p$-values.

## Bacillota/Bacteroidota ratio

The Bacillota/Bacteroidota ratio was calculated by dividing the sum of signals of HITChip phylum (level-1) data of the respective phylum. In addition, Bacillota and Bacteroidota phylum-specific qPCR assays (Suppl. Table S8) were used to quantify the respective population to calculate the Bacillota/Bacteroidota ratio. The Bacillota/Bacteroidota ratio was log-transformed (log xi + 1) for effective visualisation in the scatter plot to show the seasonal changes in the Bacillota/Bacteroidota ratio.

## Short-chain fatty acids analysis by HPLC

Faecal samples were diluted with deionised water to a 10% (w v$^{-1}$) concentration, and suspended by a bead beater using glass beads (3 mm). The suspension was centrifuged at 20,000 ×$g$ for 10 min at 4 °C, and the supernatant was passed through a 0.2 μm PTFE syringe filter (Merck). The filtrate was used for measuring the concentration of short-chain fatty acids (SCFA) using high-performance liquid chromatography (HPLC) (Accela, Thermo) using Metacarb 67H column (A5244, Agilent) at 40 °C and refractive index detector[72]. The mobile phase of 0.01 N H$_2$SO$_4$ at a flow rate of 0.6 ml min$^{-1}$ was used. The significant difference in the SCFA production between the two gut microbiota clusters and the categorised study groups was calculated by a two-sided Wilcoxon test.

## Diet and seasonal effects

As a high variability was observed in the recruited subjects, a confounder analysis using Linear Regression (IBM SPSS Statistics, Version 29.0.2.0) was carried out to find out other factors apart from the consumption of two fermented foods (*Hawaijar* and *Dahi*), by checking the variability in age, sex, BMI, habitual diet and lifestyles, and 48 h diet recall data retrieved from the questionnaire used during sample collection. The 48 variance data used for the confounder analysis are provided as Metadata in the Source Data. As most of the genus-level taxa significantly differed between diet groups and also differed by seasons (ANOVA), the Tukey post hoc test of the linear mixed-effects model was used to correct the confounders to calculate the significance of diet and the seasonal effects. In this model, we maintained the subject as a random effect, and diet and season as fixed effects. The coefficient of diet effects (with respect to the control diet, Group-A) was calculated after controlling the season and subject effects. The coefficient shows how much the diet alters the relative abundance of gut bacterial taxa, with respect to the control diet (Group-A). Similarly, the significant seasonal effects (with respect to autumn) were also calculated after controlling the diet and subject effects. The significant associations of taxa on diet or season effect were calculated by the Wilcoxon test, matched for age, BMI and gender; Benjamini-Hochberg corrected, FDR < 0.25 and absolute fold-change >2. The samples were un-paired for the comparison between four diet groups while paired for the seasonal changes. The positive and negative diet and seasonal effects were visualised by a heat map. Most of the above analyses were done using R. The changes in the similarity in gut microbiota over three seasons were calculated by comparing HITChip probe profiles of paired time interval samples using the Pearson correlation coefficient. The seasonal changes in similarity were visualised by box plots for each study group. ANOVA was performed to check the significant changes between the seasons. To visualise both the diet and seasonal effects together, principal response curve (PRC) analysis[73] was carried out using Canoco software v4.52. The changes in Canonical coefficient (Cdt) in three categorised study groups (Group-B, C and D) were plotted at three-time points along with the changes in Species weight (b$_k$) by keeping the control (Group-A) data as a baseline. The Monte Carlo Permutation procedure ($n = 499$) was used to assess the statistical significance of the variation between the categorised groups. The top 20 discriminating gut bacterial taxa or metabolites between the categorised groups (two gut microbiota clusters, fermented milk and fermented soybean impact) in the Indian study population were visualised with Random Forest (RF) analysis and plotted based on the mean decrease of accuracy (MDA) score. The RF plot was generated by using a "random forest" package with 10,000 trees in R.

## Co-occurrence network analysis

Spearman correlation coefficient ($r$) was calculated to visualise the gut bacterial co-occurrence network[51] at the genus level (level-2) relative abundances in the overall Indian study population. A combination of the R ("i-graph" package) and the 'Gephi' (Version 0.8.2-beta) (https:// gephi.org/) was used to visualise these connections. The $p$-value limit of 0.001 and r thresholds of $r \leq -0.4$ or $r \geq +0.4$ were given to visualise the modularity groups with the strongest positive correlation and co-exclusion.

## Bimodality visualisation by density plot

The unimodality score for the prevalent gut bacterial taxa (detected in >20% of the samples with a log HITChip probe signal of ≥2.8) was calculated. The scores have been calculated for 76 baseline subjects matched between European and Indian (summer season). A scatter plot of the unimodality score for the prevalent 67 taxa was used to visualise the multimodality, and a score of less than 0.5 is considered here as distinct bimodal distribution. The gut bacterial taxa with a bimodal distribution that are significantly different between European and Indian after controlling for subject effect with a linear model were visualised. The shift in the bimodal distribution of the significantly differing taxa (calculated by Tukey post hoc test of the linear mixed-effect model with false discovery rate FDR < 0.25) with diet and seasonal associations were visualised with density plots. The density plots to visualise the bimodal distribution was drawn using R. The log 10 transformed HITChip abundances at L2 taxa were used to generate the Kernel density plots.

## Statistics & reproducibility

In the study design, 78 Indian subjects with self-reported biological sex of 40 males (51.3%) and 38 females (48.7%) and a set of European individuals ($n = 78$) matched for sex, BMI, and age were used. Indian subjects were categorised into four diet groups (-20 subjects per group), and three-time points faecal samples (seasonal) were collected for the analysis. The subjects from diet groups were randomly selected from the list of subjects who qualified with the selection criteria. Sample size determination was done with Post-hoc power analysis with a power of >0.85. Six subjects were discontinued after the first sampling, and seven seasonal samples were missed. The sample details were kept blind during the experimental analysis without knowing the study sample groups. HITChip microarray analysis was performed in duplicate. As a quality threshold for microarray analysis, probe signals generated from Cy-3 and Cy-5 labelling of each sample with a Pearson correlation coefficient of 0.98 were accepted. HPLC and qPCR assays were performed in triplicate. For qPCR assays, an eight-point standard graph of 10$^1$-10$^8$ gene copies with assay efficiency of more than 0.99 was used for absolute load quantification of bacterial taxa. For this, the DNA extracted from the microbial strains of *Escherichia coli* DH5α, *Bacteroides thetaiotaomicron* DSM 2079, *Ruminococcus gnavus* ATCC 29149, *Methanobrevibacter smithii* DSM 274, *Ligilactobacillus plantarum* WCFS1, *Bacillus subtilis* ATCC6051, *Prevotella copri* DSM-18205, *Bifidobacterium longum* subsp. *longum* DSM 20219, and *Bacteroides fragilis* were used as standard. A mixture of short-chain fatty acids with known concentrations was used as internal standards during HPLC analysis. For LC-HRMS analysis, Tarocholate was used as an internal standard.

Permutation multivariate analysis of variance (PERMANOVA) test with 9999 permutations using Bray-Curtis dissimilarity matrices was used to show the statistical significance in the overall difference in the bacterial community composition or metabolites profile of the study groups using PAST v3.22[70]. The supervised redundancy analysis (RDA) was performed using Canoco software v4.52 with Montecarlo permutation test ($n = 499$) for finding a significant association of taxa or metabolite with the diet and lifestyle metadata. A two-sided Wilcoxon rank sum test was carried out to compare individual taxa or metabolites between the groups, and BH corrected $p$-values using R version 3.1.3. A two-sided Wilcoxon test with Bonferroni correction was used to show the statistical significance between the diet or season groups. The Principal Response Curve (PRC) was carried out using Monte Carlo

Permutation ($n$ = 499) to assess the relevance of the effect of both diets and seasons together (Canoco v4.52). A confounder analysis using Linear Regression (SPSS 29.0.2.0) was done to determine the significant impact of other factors apart from consuming the two fermented foods using the metadata retrieved from the questionnaire. The Tukey post hoc test of the linear mixed effect model with false discovery rate (FDR < 0.25) was used to correct the confounders and assess taxa with significant diet and seasonal association. Descriptive statistics for continuous variables with normal distribution were presented as mean ±s.d and mean ± s.e.m. A log transformation was applied when the distribution deviated from the normality.

### Reporting summary

Further information on research design is available in the Nature Portfolio Reporting Summary linked to this article.

## Data availability

Data sets used in this study, including the probe-level Human Intestinal Tract phylogenetic microarray (HITChip) data, the associated sample metadata and the derived higher-level genus and phylum level taxonomic abundance tables, qPCR data on absolute abundance and chemical profiling data are available with a permanent DOI via Zenodo (https://doi.org/10.5281/zenodo.14424024) with MIT license. The data is part of the HITChip Atlas (with project title "CREST STUDY"), maintained by the Laboratory of Microbiology, Wageningen University & Research. The microbiota profiling data for the fermented foods (16S rRNA gene amplicon sequencing data) is available in NCBI-SRA database (accession number: PRJNA1191989). The description of the source data used to make the figures is provided in an Excel file in this manuscript. The individual-level metadata on the diet and lifestyle habits of individuals are available in source data and supplementary data. The processed data generated in this study are provided in the Supplementary Information. Source data are provided with this paper.

## Code availability

The code used to perform the analyses and generate results in this study is openly available with an MIT license and has been deposited in Zenodo at https://doi.org/10.5281/zenodo.14369940[74].

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

## Acknowledgements

This research was supported by the Department of Biotechnology (DBT), Ministry of Science and Technology, Government of India through the DBT-CREST award (Department of Biotechnology Cutting-edge Research Enhancement and Scientific Training Award)

to K.J. (BT/IN/CREST-Awards/44/KJ/2010-11). The financial support of the Institute of Bioresources and Sustainable Development (IBSD), India, to K.J. is also gratefully acknowledged. Spinoza's grant of WMdV for the HITChip analyses and the Research Council of Finland (decisions 295741, 330887) support to LL are gratefully acknowledged. We acknowledge Padma Ramakrishnan for supporting the identification of the faecal metabolites by MS/MS spectra analysis, Ngangyola Tuikhar and Rajendra Kumar Labala for supporting R scripts and statistical analysis, and Santosh Keisam for supporting the fermented food microbiota analysis. We are also indebted to Thangjam Anand Singh, Wahengbam Romi, Khunjamayum Romapati Devi, Lourembam Lolen Singh, Khundrakpam Basanti, Santosh Keisam, Kennedy Singh Karam, and Hanglem Nabadwip Singh for helping with the sample collection; Phillipe Puylaert, Susana Fuentes, Gerben Hermes, Floor Hugenholtz, Wilma Akkermans-van Vliet and Ineke Heikamp-de Jong from the Laboratory of Microbiology, Wageningen University for their support of this work. We are grateful to the members of our study groups for their wholehearted support in providing three seasonal faecal samples. IBSD communication number is IBSD/2020/01/064.

## Author contributions

The project idea was conceived, and the work was designed by K.J., E.G.Z., and H.S. K.J., H.S., E.G.Z., and W.Md.V. managed the project. K.J. performed sample collection and experiments. H.H. and T.V.G. supported the experiments. K.J., L.L., S.T., E.G.Z., H.S., and W.Md.V. designed the data analysis, and K.J., L.L., and S.T. performed data analysis. K.J. wrote the paper. L.L. made the code/data available. E.G.Z., H.S., W.Md.V., L.L. revised the manuscript.

## Competing interests

The authors declare no competing interests.
