## [Transparent Peer Review file · Nature Communications]

Fermented foods affect the seasonal stability of gut bacteria in an Indian rural population

Corresponding Author: Dr Kumaraswamy Jeyaram

Version 0:

Reviewer comments:

Reviewer #1

(Remarks to the Author)

the work by Jeyaram et al is an observational study carried out on an Indian agrarian population, evaluating differences in gut microbiota composition and metabolites according to the consumption of two typical fermented foods. Although the study is interesting in principle and try to address an important question (the impact of fermented foods on gut microbiota and health), according to the reviewer there are several limitations in the methodology that limit its validity and the conclusions drawn.

MAIN ISSUES:

-the sample size seems too low. did you carried out a sample size calculation?
in addition, high variability in the recruited subjects (age, sex, BMI). the differences observed between the groups may be due to other factors apart from the consumption of the two specific fermented foods.

-No data on the diet is provided, but the 4 groups may also differ for habitual diet.
for example, is there the consumption of OTHER fermented foods (apart from the 2 considered)?

-Use of microarray. this technology is quite obsolete and has been replaced by other techniques (e.g. shotgun metagenomics)... therefore the results obtained are not comparable with other studies already present in literature, as well as future works.

-where microbial loads of Bacillus and LAB come from? which variability is reported in literature? since these products are mainly home-made, I'd expect high variability in microbial composition (for sure, not only Bacillus/LAB are present), as well as in manufacturing.

-Microbiota analysis of the fermented products would be also necessary to support the results.

other comments:

Methods:

How did you define sample size? is there any power analysis?

in addition, high variability In the study population (within and between groups) is observed e.g., in age, BMI.

did you corrected for confounding factors in statistical analysis? is it still significant after correction for age for example?

Also data on the diet should be provided and any differences in diets among the groups should be reported, since different diets may be an important confounding factor.

details on randomization should be provided, e.g., software used, variables considered.

Also, details on the number of subjects screened, n. of subjects enrolled, drop-outs should be detailed.

Sample collection: more details should be provided (e.g., kit used, use of anaerobiosis, refrigerated transport?). did you follow any SOP (e.g., IHMS SOPs for sample collection and storage <https://human-microbiome.org>)

Introduction: some important aspects should be considered and papers in the field are missing, e.g., the importance of microbial intake as stated by ISAAP <https://www.sciencedirect.com/science/article/pii/S0022316623126228>

also the aspect relative to the possibility of microbes transfer from foods to gut microbiome is not considered e.g. <https://www.nature.com/articles/s41467-020-16438-8>

Data availability:

Data on the single subjects are not available, only average data for each group. a supplementary dataset with info on each subject (age, sex, BMI etc) should be made available.

In addition, also data retrieved from dietary questionnaires. it is stated that they did 48-h recall, but no data is presented.

Figure 3B and supplementary Figure S2: the use of matched line-plots should be avoided, since you do not have matched samples (e.g., same subject at 2 time-points). how did you decide how subjects were matched with each other?

Reviewer #2

(Remarks to the Author)

This is an interesting study that investigated the effect of regular consumption of fermented foods on the gut microbiota of a rural population in India. The study is well executed and uses appropriate methodologies. The study highlighted seasonal shifts the gut microbiota, in particular in the Bacteroidota community and associated changes in the faecal metabolome with particular reference to a number of fatty acid derivatives. Regular consumption of fermented foods was suggested to result in reduced gut microbiota diversity and reduced bacterial loads.

A number of observations relating to the data were made that should be addressed:

Line 153: Check to confirm if 20 subjects should read 19 subjects?

Lines 159-161 and Figure 1: While Cluster-P and Cluster-B/R are defined at lines 159-161, Cluster 1 and 2 are used in Figure 1. Suggest using Cluster-P and Cluster-B/R throughout and modifying Figure 1 accordingly.

Figure 1 A: The quality of the image displaying the HITChip-Oligo profile data is very poor. Can a higher resolution image be included. It may also be helpful if this image was expanded and displayed as a single image in the Supplementary Information.

Figure 2: This figure seems to be constructed using a smaller number of data points to Figure 1. Is this correct and if so explain why in the text of the manuscript. What impact does this have in the overall data analysis and conclusions?

Figure 2B and Table S3: 22 faecal metabolites were used to construct Figure 2B whereas 23 faecal metabolites that differ significantly are listed in Table S3. The additional metabolites in Table S3 appears to be compound ID C1806. If 23 compounds should be included in the analysis, correct in the manuscript and describe any subsequent impact on the conclusions from the data analysis. If 22 compounds only should be included in the analysis explain to the Editor why 23 are included in Table S3 and correct Table S3 accordingly.

Lines 232-233 and Figure S1 B: Suggest that the text should read “(~20 mean decrease accuracy)” as oppose to “(> 20); as the data in Figure S1 B indicates that the Mean decrease accuracy score for C1806 is ~ 20 while that for C1767 is ~ 19.5.

Lines 253-258 and Figure 3A: Figure 3A is a PCA plot of European (pink dots) and Indian (blue/green dots) which separates the two groups into two clusters in particular relative to Comp. 1. However, in the text it is stated that the separation was “associated with the Prevotella status, irrespective of Indian or European subjects”. Please review this section to ensure there is no error, and if not, edit the text to better explain the observations from the data.

While the manuscript is well written, a number of minor edits for grammar and style are suggested. Many (but not all) of these relate to the Introduction. When addressed, the manuscript would benefit from a final review to specifically focusing on grammar and style.

Line 63-64: Suggest moving this statement about health claims and inserting it immediately before the sentence starting on line 60 that refers to validated claims. This edit is suggested as it builds the case better to first mention “claims” followed by “validated claims”.

Line 66: Delete “predominantly with”

Line 68: Suggest editing to “of the phylum Bacillota, in addition to yeasts and fungi”

Line 75: Suggest editing to “whereas genetics is likely not a valid driver”

Line 78: Change “grouped” to “group”

Line 79: Insert “and” following “Bacteroides entrotypes 1 and 2,”

Line 80-81: Replace “the traditional population” with “human populations that follow a traditional lifestyle” and delete “the modern industrialised population” and replace with “populations exposed to modern industrialised lifestyles”

Line 82: Suggest deleting “stable” at “to be stable due to”

Line 84: Suggest deleting “on” and replacing with “the prevalence”

Line 85: Suggest inserting “specific” following “individual”. Change “return” to “returning”

Line 89: Insert “changes in the gut microbiota populations are cyclic (annual)” Following “hunter gathers”

Line 94: Start the sentence with “Examples include” and edit remainder of sentence to ensure correct grammar

Line 103: This sentence needs to be edited with particular reference to explaining “eliminate immediately” and “shed more

days”

Line 107: Delete “by” and replace with “following consumption of”

Line 111: Delete “for” and replace with “using”

Line 114: Insert “the prevalence of” before “Prevotellaceae”

Line 117: Insert “of the” following “two”

Line 124: Replace “lives” with “living”

Line 126: Insert “were” following “groups”

Line 127: The objective of the study is described as the faecal bacteria, while the manuscript title and the Introduction refer to “gut” bacteria. Suggest qualifying the statement by highlighting that the faecal bacteria are commonly used as a marker for the large intestine i.e. colon microbiota populations.

Line 153: Should “and” be replaced by “or”?

Line 236: There is reference to acetate and propionate production but no associated data is presented. This should be highlighted as “(data not shown)”. The significance/relevance of the statement about acetate and propionate should also be included.

Line 223: Sentence starting with “The MS/MS” does not read well/correctly, something is missing

Line 399: This should read “did not differ” as oppose to “not differed” as currently in the text

Line 445: Suggested editing “were also significantly varying over seasons” to “also varied significantly over season”

Line 548: Suggest editing to “... .. visible in traditional populations living in uncontrolled household environments,”.

Also, what do the author mean by “uncontrolled”?

Line 555-557: Meaning/context of the statement “important concerning seasonal patterns” is unclear

Line 572-573: Sentence is poorly constructed

Line 587: Suggest editing to “... prevent pathogen establishment,”

Line 608: Suggest editing to “... produced it via the butyrate”

Line 660: HDI should be in brackets i.e., (HDI)

Line 694: Suggest inserting “the” immediately before “above”

Lines 695-696: Change “consuming” to “consumed” throughout this sentence

Version 1:

Reviewer comments:

Reviewer #1

(Remarks to the Author)

Although I still have some concerns about the small sample size (20 subj x group) and the obsolete methodology used, the authors have addressed all the other points raised.

(Remarks on code availability)

Reviewer #2

(Remarks to the Author)

That authors have addressed the individual comments raised from review of the original manuscript in an adequate manner. A number of errors relating to the data or the way it was presented in the original article were acknowledged and addressed, while a number of suggested edits relating to style and grammar were amended.

(Remarks on code availability)

could not find where to access code

Detailed Responses to the Reviewers' Comments

REVIEWER #1:

COMMENT 1. The work by Jeyaram et al is an observational study carried out on an Indian agrarian population, evaluating differences in gut microbiota composition and metabolites according to the consumption of two typical fermented foods.

Although the study is interesting in principle and try to address an important question (the impact of fermented foods on gut microbiota and health), according to the reviewer there are several limitations in the methodology that limit its validity and the conclusions drawn.

Response: Thank you for the positive remark and critical comments on improving the manuscript. We have done some additional statistical analysis and provided additional data, as suggested, to support the validity of the results drawn from the study. We have addressed all queries and mentioned the limitations in the revised manuscript. Please see below the detailed responses to the reviewer's comments.

COMMENT 2:

The sample size seems too low. Did you carry out a sample size calculation?

Response:

We agree that the sample size is relatively low, 214 seasonal samples from 79 Indian subjects, and samples of 78 European subjects for comparison. However, it is important to note that only a limited number of studies address the gut microbiota of people from India. Moreover, it is not trivial to obtain samples from the Indian subjects that consume the studied fermented foods.

Finally, there were considerable legal challenges to overcome in transferring the faecal samples from India to The Netherlands, where the analysis was done during the post-doc stay of the first

author. We surveyed with a questionnaire on the consumption of the two fermented foods among thousands of people in the study population to fix ~20 subjects each in four diet groups. The main reason for this sample size limitation is finding proper control subjects (people never consumed the targeted two fermented foods at least for the last ten years). Moreover, the persons with a high frequency (3 times/ week to daily) in consumption of the above-fermented foods were recruited for the study.

We did a post-hoc power analysis for sample size calculation using <https://homepage.univie.ac.at/robin.ristl/samplesize.php?test=ttest>, in assumptions based on the gut microbiota composition of published Indian populations by assessing the power of detection of 2-fold difference (mean difference= 2) at 0.05 significance level in the standard deviation of major bacterial taxonomic group of interest, which resulted power of > 0.85 with a 20 sample size per group.

As per the Editor's suggestion (Please see the Editor's comment #2 and response), we have acknowledged that the relatively small sample size is a limitation of our work and mentioned in the Discussion section concluding remarks, in line number 738-742, page 34.

COMMENT 3:

In addition, high variability in the recruited subjects (age, sex, BMI), the differences observed between the groups may be due to other factors apart from the consumption of the two specific fermented foods.

-No data on the diet is provided, but the 4 groups may also differ for habitual diet.

for example, is there the consumption of OTHER fermented foods (apart from the 2 considered)?

Response:

We have provided the metadata on variability in the recruited individual subjects, habitual diet and lifestyles as a separate supplementary Excel file with three sheets containing the following information in the revision.

Sheet 1: Supplementary File S3. Differences in the age, sex, BMI, clan, nature of birth and marital status of individual subjects in each categorised study group.

Sheet 2: Supplementary File S4. Differences in the long-term habitual diets and lifestyles of individual subjects in each categorised study group.

Sheet 3: Supplementary File S5. Differences in the 48h diet recall data during seasonal sampling and associated bacterial community clusters in individual subjects in each categorised study group.

We have checked the confounders by Linear Regression analysis using SPSS software to know other factors (apart from the consumption of the two specific fermented foods) on the observed difference between the diet groups (Group A, B, C, D), and the bacterial community structure based two clusters (Cluster-P and Cluster-B/R). The variability (47 variances) in the recruited subjects (age, sex, BMI), habitual diet and lifestyles, and 48 h diet recall data were used for the confounder analysis. Please see the response to Reviewer #1 Comment 8, for the details.

Regarding the consumption of other fermented foods (apart from the 2 considered in the study), the study population consumed *Soibum* & *Soidon* (fermented bamboo shoots) and *Hentak* (fermented fish paste) with very little frequency, which were cooked before consumption. The 48h diet recall questionnaire data showed less consumption frequency of *Soibum* (24/214),

Soidon (3/214), and *Hentak* (3/214) in the study population. The two fermented foods considered in this study (*Hawaijar* and *Dahi*) are consumed mostly in fresh form with live microbes, and subjects consuming more frequently (> 3 times/ week to daily) were considered for this study to know the effect. Confounder analysis did not relate the difference visualised in the diet groups with the *Soibum*, *Soidon* or *Hentak*.

Mentioned in the line number 496-502, page 24.

COMMENT 4:

-Use of microarray. this technology is quite obsolete and has been replaced by other techniques (e.g. shotgun metagenomics)... therefore the results obtained are not comparable with other studies already present in literature, as well as future works.

Response:

We do agree that shotgun metagenomics is the present-day choice for gut microbiota studies. The results obtained from the microarray (HITChip) used in our study are also comparable with next-generation sequencing-based microbial community analysis. HITChip hybridisations and resulting community profiles correlate very well with pyrosequencing-based microbial profiles in earlier studies and Illumina metagenome studies in later works. Please see the references of Claesson et al, 2009; van den Bogert et al. 2011, Arumugam et al., 2011, Le Chatelier et al., 2013 listed here. Therefore, equivalent biological conclusions are obtained. Of note, the HITChip database, consisting of thousands of microbiota profiles, that was used in our study to compare and contrast the Indian and European datasets has the unique feature that all samples are analysed with identical protocols which, in contrast to the continuously changing sequencing protocols, allows a one-to-one comparison between samples from different projects.

References:

- Claesson MJ, O'Sullivan O, Wang Q, et al. Comparative analysis of pyrosequencing and a phylogenetic microarray for exploring microbial community structures in the human distal intestine. *PLoS One*. 2009;4(8):e6669. Published 2009 Aug 20.
doi:10.1371/journal.pone.0006669
- van den Bogert B, de Vos WM, Zoetendal EG, Kleerebezem M. Microarray analysis and barcoded pyrosequencing provide consistent microbial profiles depending on the source of human intestinal samples. *Appl Environ Microbiol*. 2011 Mar;77(6):2071-80. doi: 10.1128/AEM.02477-10. Epub 2011 Jan 21. PMID: 21257804; PMCID: PMC3067328.
- M Arumugam, J Raes, E Pelletier, D Le Paslier, T Yamada, DR Mende, GR Fernandes, J Tap, T Bruls, JM Batto, M Bertalan, N Borrueal, F Casellas, L Fernandez, L Gautier, T Hansen, M Hattori, T Hayashi, M Kleerebezem, K Kurokawa, M Leclerc, F Levenez, C Manichanh, HB Nielsen, T Nielsen, N Pons, J Poulain, J Qin, T Sicheritz-Ponten, S Tims, D Torrents, E Ugarte, EG. Zoetendal, J Wang, F Guarner, O Pedersen, WM de Vos, S Brunak, J Doré, MetaHIT Consortium, J Weissenbach, SD Ehrlich, and P Bork (2011) Enterotypes of the human gut microbiome. *Nature* 473: 174-80.
- Le Chatelier E, Nielsen T, Qin J, Prifti E, Hildebrand F, Falony G, Almeida M, Arumugam M, Batto JM, Kennedy S, Leonard P, Li J, Burgdorf K, Grarup N, Jørgensen T, Brandslund I, Nielsen HB, Juncker AS, Bertalan M, Levenez F, Pons N, Rasmussen S, Sunagawa S, Tap J, Tims S, Zoetendal EG, Brunak S, Clément K, Doré J, Kleerebezem M, Kristiansen K, Renault P, Sicheritz-Ponten T, de Vos WM, Zucker JD, Raes J, Hansen T; MetaHIT consortium, Bork P, Wang J, Ehrlich SD, Pedersen O (2013) Richness of human gut microbiome correlates with metabolic markers. *Nature* 500:541-6.

Due to technical limitations, we did not use shotgun metagenome sequencing when the project was initiated. However, we now have implemented the amplicon sequencing-based microbial community profiling by using Illumina-MiSeq 16S rRNA amplicon sequencing for the microbiota analysis and used this for analysing the two fermented foods in which the effect was studied here. The microbiota data of the two fermented foods and MG-RAST sequence data accession numbers are included in the supplementary information (Table S7 and S8, pages 37-40) in the revised manuscript, and please see for the details to the response to Reviewer#1 Comment 5.

Moreover, we also used microbial taxa-specific qPCR assay to support the results obtained from microarray experiments. Therefore, the results obtained in the present study are comparable with those already present in the literature and for future works.

COMMENT 5:

-where microbial loads of *Bacillus* and LAB come from? which variability is reported in literature? since these products are mainly home-made, I'd expect high variability in microbial composition (for sure, not only *Bacillus*/LAB are present), as well as in manufacturing.

-Microbiota analysis of the fermented products would be also necessary to support the results.

Response:

To understand the effect of long-term consumption of fermented foods (with live bacteria) on gut bacteria, two home-made fermented foods, a fermented soybean product *Hawaijar* (dominated by *Bacillus*) and a fermented milk product *Dahi* (dominated by lactic acid bacteria) were targeted in this study.

In healthy adult faeces, the load of lactic acid bacteria accounts for up to 10^6 – 10^8 CFU/g (Mitsuoka et al., 1996; Walter 2001; Mikelsaar 2016; De Filippis et al., 2020) and *Bacillus* spp. in the range of 10^3 - 10^8 CFU/g (Tam et al. 2006) which represented a small fraction (< 0.01% relative abundance) of the total faecal bacterial load of up to 10^{12} CFU/g (O’Hara and Shanahan 2006). In our study, the faecal samples contained the lactic acid bacteria load of 10^6 - 10^{11} CFU/g and the *Bacillus* load of 10^4 - 10^9 CFU/g with a total bacterial load ranging 10^{10} - 10^{12} CFU/ g, quantified by qPCR assays. Mentioned in the Line 442-448, page 22.

As suggested, we assessed the bacterial composition of two fermented foods (*Hawaijar* and *Dahi*, 5 samples each) by 16S rRNA gene amplicon sequencing (Illumina- MiSeq). We have included the data with MG-RAST accession number for the bacterial composition variability at Phylum, Family and genus levels in Supplementary Tables S7 and S8 (Pages 38-41). The methodology used for this is mentioned in Supplementary Methodology, 1.2 Page 7, Lines 151-172. Further quantification by qPCR assay, resulted in the dominant presence of lactic acid bacteria at 10^8 - 10^9 CFU/ g in the fermented milk *Dahi* and *Bacillus* of 10^{10} - 10^{11} CFU/g in fermented soybean *Hawaijar*. This quantification of Lactic Acid Bacteria and *Bacillus* aims to understand the changes in the gut microbiota due to long-term frequent intake of live bacteria through fermented foods. The results are presented in Supplementary Figure S7 (Page 16).

References:

Mitsuoka T. The Human Gastrointestinal Tract. In: Wood BJB (ed.). The Lactic Acid Bacteria Volume 1: The Lactic Acid Bacteria in Health and Disease. Boston, MA: Springer US, 1992, 69–114.https://doi.org/10.1007/978-1-4615-3522-5_4.

- Walter J, Hertel C, Tannock GW et al. Detection of *Lactobacillus*, *Pediococcus*, *Leuconostoc*, and *Weissella* species in human feces by using group-specific PCR primers and denaturing gradient gel electrophoresis. *Appl Environ Microbiol* 2001;67: 2578–85. <https://doi.org/10.1128/AEM.67.6.2578-2585.2001>
- De Filippis, F., Pasolli, E. and Ercolini, D., 2020. The food-gut axis: lactic acid bacteria and their link to food, the gut microbiome and human health. *FEMS microbiology reviews*, 44(4), pp.454-489. <https://doi.org/10.1093/femsre/fuaa015>
- O'Hara AM, Shanahan F. The gut flora as a forgotten organ. *EMBO Rep* 2006;7:688–93. <https://doi.org/10.1038/sj.embor.7400731>
- Mikelsaar, M., Sepp, E., Štšepetova, J., Songisepp, E. and Mändar, R., 2016. Biodiversity of intestinal lactic acid bacteria in the healthy population. *Advances in Microbiology, Infectious Diseases and Public Health: Volume 4*, pp.1-64. https://doi.org/10.1007/5584_2016_3.
- Tam NK, Uyen NQ, Hong HA, Duc le H, Hoa TT, Serra CR, Henriques AO, Cutting SM. The intestinal life cycle of *Bacillus subtilis* and close relatives. *J Bacteriol.* 2006 Apr;188(7):2692-700. <https://doi.org/10.1128/jb.188.7.2692-2700.2006>

COMMENT 7:

other comments:

Methods: How did you define sample size? is there any power analysis?

Response:

We used post-hoc power analysis for sample size calculation using <https://homepage.univie.ac.at/robin.ristl/samplesize.php?test=ttest>, in assumptions based on the gut microbiota composition of published Indian populations, by assessing the power of detection

of 2-fold difference (mean difference= 2) at 0.05 significance level in the standard deviation of Shannon diversity and the abundance of major bacterial taxonomic groups of interest like *Prevotella*, *Akkermansia* in gut microbiota , with a group sample size of 20 subjects, yielded a power of > 0.85. We agree that the sample size is low, with 214 samples from 79 Indian study populations (four groups, ~20 subjects each) and 78 European subjects samples for comparison. Please see the Editor's suggestion (Editor's comment #2), as our study cannot be strengthened with additional samples; we clearly acknowledged that the relatively small sample size is a limitation of our work in the Discussion section concluding remarks, in lines number 738-742, Page 34. We have mentioned this analysis in the method section of line number 767-768, page 35

COMMENT 8

in addition, high variability In the study population (within and between groups) is observed e.g., in age, BMI. Did you correct for confounding factors in statistical analysis? Is it still significant after correction for age for example? Also, data on the diet should be provided and any differences in diets among the groups should be reported, since different diets may be an important confounding factor. Details on randomisation should be provided, e.g., software used, and variables considered.

Response:

We have done confounder analysis by Linear Regression (IBM SPSS Statistics, Version 29.0.2.0) to find out other significant factors apart from consumption of two fermented foods and other significant drivers for the two gut bacterial community clusters (Cluster-P and Cluster-B/P) visualised in the study population, by checking the variability in the recruited subjects (age, sex,

BMI), habitual diet and lifestyles, and 48 h diet recall data retrieved from the questionnaire used during sample collection. Forty-seven variance data of individual subjects were used for the confounder analysis and are provided as additional Supplementary Excel file (Supplementary files S3-S5).

We have corrected the confounding factors with a Linear mixed-effect model. We controlled the season and subject effects (age, sex, BMI) to visualise the significant diet effects, and controlled the diet and subject effects to visualise the seasonal effect. The diet effects after controlling the confounding factors (based on the Tukey post hoc test of the linear mixed-effect model), and the bacterial taxa with significant diet association after correction of confounders (False discovery rate $FDR < 0.25$ adjusted p-value, BH corrected) are visualised. The effect of the season is still significant after the correction for *Prevotella tannerae* et rel., *Parabacteroides distasonis* et rel., *Tannerella* et rel., and *Bacteroides splanchnicus* et rel., not for *Catenibacterium mitsuokai* et rel. (mentioned in lines 481-483, Page 23).

The significant confounders observed between the two community clusters (Cluster-P and Cluster B/R) are included in the Result section, in line numbers 177-190, Pages 7-8.

“We did a confounder analysis using Linear Regression (SPSS Statistics) to find out other significant factors driving these two bacterial community clusters by checking the variability in the recruited subjects (age, sex, BMI), habitual diet and lifestyles, and 48 h diet recall data retrieved from a questionnaire used during sample collection. The variability in metadata on diet and lifestyle habits (47 variances) of individual subjects recruited for the study is provided as a Supplementary Excel file (Supplementary files S3-S5). The cofounder analysis resulted in a significant negative association of BMI with *Prevotella* (standardised regression coefficient $\beta = -$

0.287, $p=0.009$). BMI showed a subtle but significant difference between the two clusters, Cluster-P with a median BMI of 23.65 and Cluster-B/R with a median BMI of 25.65 ($p=0.0021$, T-test, two-tailed, unequal variances). In addition, we observed a significant negative association of smoking with the *Clostridium* cluster-XVIa ($\beta= -0.381$, $p=0.0047$) and the long-term-dietary habit of duck meat consumption with *Bifidobacterium* ($\beta=-0.402$, $p=0.009$). However, smoking and duck consumption did not differ significantly between Cluster-P and Cluster-B/R.”

In lines 254-256, Page 11

“The confounder analysis with dietary habits resulted in a significant positive association of butyrate with pork consumption ($\beta= +0.255$, $p=0.013$). However, pork consumption did not differ significantly between the Cluster-P and Cluster-B/R.”

In lines 467-476, Page 23

“The diet groups (A, B, C, D) did not differ significantly in terms of sex (Fisher exact test, $p=0.08$) or age (Kruskal-Wallis test $p=0.76$), but differed significantly with BMI (Kruskal-Wallis test $p=0.007$). As a high variability in the dietary habits recorded in individual subjects and seasonal variation in gut microbiota, we corrected the confounding factors by a Linear mixed effect model. We controlled the season and subject effects (age, sex, BMI) to visualise the significant diet effects (Group A-D) and the diet and subject effects to visualise the seasonal effects. For this purpose, we used the Tukey post hoc test of the linear mixed model with a False discovery rate. The diet effects after controlling the confounding factors, the bacterial taxa with significant diet association (False discovery rate $FDR<0.25$ adjusted p -value, BH corrected) are visualised in Fig. 7A and Suppl. Table S9.”

The model, software and variance used for finding the confounder and correcting the confounders are included in the Method Section, Lines 905-911, Page 41.

“As a high variability was observed in the recruited subjects, a confounder analysis using Linear Regression (IBM SPSS Statistics, Version 29.0.2.0) was carried out to find out other factors apart from the consumption of two fermented foods (*Hawaijar* and *Dahi*), by checking the variability in age, sex, BMI, habitual diet and lifestyles, and 48 h diet recall data collected using a questioner during sample collection. The variance data (47 variances) used for the confounder analysis are provided as a Supplementary Excel file (Supplementary files S2-S4). As most of the genus-level taxa significantly differed between diet groups and also differed by seasons (ANOVA analysis), the Tukey post hoc test of the linear mixed-effects model was used to correct the confounders to calculate the significance of diet and the seasonal effects.”

The subjects of four diet groups were randomly selected from the list of subjects qualified with the selection criteria, double blind randomisation was used for the selection. The faecal sample DNA/ metabolite extraction was kept blind without knowing the sample group in the study. Mentioned in line 768-770, Pages 35-36.

The details of other software used, randomisation, and variables considered are listed below and mentioned in the method section (Lines 788-961, pages 36-43).

The supervised redundancy analysis (RDA) was performed using Canoco software v4.52 (Wageningen University, The Netherlands) with the Monte-Carlo permutation test (n=499) for finding the significance of the gut bacteria association with metadata.

Multivariate analysis PERMANOVA test was performed using PAST v3.22 software with 9999 permutations.

The principal response curve (PRC) analysis was carried out to visualise both the diet and seasonal effects together using Canoco software v4.52 (Wageningen University, The Netherlands) with the Monte Carlo Permutation procedure (n=499).

Random Forest (RF) analysis was done by using a "random forest" R package with 10,000 trees. Significant diet and seasonal associations were corrected by using the Tukey posthoc test of the linear mixed-effect model with a false discovery rate $FDR < 0.25$).

COMMENT 9

Also, details on the number of subjects screened, n. of subjects enrolled, drop-outs should be detailed.

Response:

We surveyed with a questionnaire on the consumption frequency and quantity of selected two fermented foods (*Hawaijar* and *Dahi*), and collected information from 1104 people. Initial screening was done with the eligibility criteria of good general health without any chronic illness, not taking antibiotics within six months, normal bowel frequency, and free from any gastrointestinal disorders. The persons who consume more than 3 times/ week to daily the above-fermented foods and not consuming at least for the last ten years were categorised. With the above criteria, ~20 subjects each in four diet groups were targeted and a total of 85 subjects were enrolled for the study, 6 subjects dropped out (2 declined during the sample collection and 4 experienced health issues and took medical treatment during sampling), and 7 subjects declined to seasonal samples (after giving the first samples). The samples from 79 subjects out of 85

enrolled were considered for the study. The details are included in Supplementary Information “Subject enrollment details” in lines 101-118, Page 5.

COMMENT 10

Sample collection: more details should be provided (e.g., kit used, use of anaerobiosis, refrigerated transport?). Did you follow any SOP (e.g., IHMS SOPs for sample collection and storage <https://human-microbiome.org>)

Response:

We have followed IHMS Standard procedures (IHMS SOP 004) for the sample collection by using Sterile Clinicol (Himedia) of 25 ml capacity sterile plastic tubes, screw-capped, wide-mouthed with a spoon. The self-collected faecal samples were frozen immediately by using a prechilled gel pack or dry ice in a cooler box and transported to the laboratory within two hours of collection and stored at -80° C and extracted the DNA within 7 days of sample collection. As the work plan focused on analysis without culturing the microbes, we did not use an anaerobiosis kit during the sample collection.

We have provided more details about the sample collection in Lines 774-779, Page 36.

COMMENT 11

Introduction: some important aspects should be considered and papers in the field are missing, e.g., the importance of microbial intake as stated by ISAAP

<https://www.sciencedirect.com/science/article/pii/S0022316623126228>

also the aspect relative to the possibility of microbes transfer from foods to gut microbiome is not considered e.g. <https://www.nature.com/articles/s41467-020-16438-8>

Response:

Thank you for suggesting two important aspects that are very much relevant to the manuscript.

We have included the first reference as “The ISAAP-sponsored National Health and Nutrition Examination Survey (NHANES) analysis witnessed positive health outcomes associated with the intake of foods with live microbes including fermented foods, as it was associated with a reduction in systolic blood pressure and an increase in HDL, which are known to decrease the risk of cardiovascular diseases.” in lines 101-105, pages 4-5. The cited reference is included in lines 1089-1091, Page 48 as follows

“22. Hill, C. et al. Positive health outcomes associated with live microbe intake from foods, including fermented foods, assessed using NHANES database. *J. Nutr.* 153, 1143-1149 (2023).”

We have included the second reference as “However, a large-scale metagenome-assembled genomes (MAGs) analysis provides unprecedented evidence that fermented foods can indeed be the primary source of lactic acid bacteria in the human gut.” in lines 115-117, Page 5. The cited reference is included in lines 1100-1101, page 48 as follows

“26. Pasolli, E. et al. Large-scale genome-wide analysis links lactic acid bacteria from food with the gut microbiome. *Nat. Commun.* 11, 2610 (2020).”

COMMENT 12:

Data availability:

Data on the single subjects are not available, only average data for each group. A supplementary dataset with info on each subject (age, sex, BMI etc) should be made available.

In addition, also data retrieved from dietary questionnaires. It is stated that they did 48-h recall, but no data is presented.

Response:

We have provided the metadata of variability retrieved from dietary questionnaires in the recruited individual subjects, as well as diet and lifestyle habits, as a separate supplementary Excel file with three sheets containing the following information in the revision.

Sheet 1: Supplementary File S3. Differences in the age, sex, BMI, clan, nature of birth and marital status of individual subjects in each categorised study group.

Sheet 2: Supplementary File S5. Differences in the long-term habitual diets and lifestyle habits of individual subjects in each categorised study group.

Sheet 3: Supplementary File S4. Differences in the 48h diet recall data during seasonal sampling in individual subjects in each categorised study group.

We have submitted this data in the public repository [10.6084/m9.figshare.26362063](https://doi.org/10.6084/m9.figshare.26362063).

COMMENT 13:

Figure 3B and Supplementary Figure S2: the use of matched line-plots should be avoided, since you do not have matched samples (e.g., same subject at 2 time-points). How did you decide how subjects were matched with each other?

Response:

Earlier, we matched the samples for age, BMI and gender between Europeans and Indians at baseline (summer), so we decided to use matched-line plots. Now, we realise that such matched line plots should be avoided. We have redrawn violin plots without line-matching in revision Fig. 3A and Supplementary Figure S3 (in revision).

REVIEWER #2

COMMENT 1:

This is an interesting study that investigated the effect of regular consumption of fermented foods on the gut microbiota of a rural population in India. The study is well executed and uses appropriate methodologies. The study highlighted seasonal shifts the gut microbiota, in particular in the Bacteroidota community and associated changes in the faecal metabolome with particular reference to a number of fatty acid derivatives. Regular consumption of fermented foods was suggested to result in reduced gut microbiota diversity and reduced bacterial loads.

Response:

Thank you very much for the positive remarks and valuable comments/suggestions for improving the manuscript.

COMMENT 2:

A number of observations relating to the data were made that should be addressed:

Line 153: Check to confirm if 20 subjects should read 19 subjects?

Response:

Thank you for pointing out the mistake. It should be 19, not 20.

Please see the correction on Page 7, line 162.

COMMENT 3:

Lines 159-161 and Figure 1: While Cluster-P and Cluster-B/R are defined at lines 159-161, Cluster 1 and 2 are used in Figure 1. Suggest using Cluster-P and Cluster-B/R throughout and modifying Figure 1 accordingly.

Response:

Thank you for the suggestion. We have corrected the terms “Cluster-1 and Cluster-2” as “Cluster-P and Cluster-B/R” in Figure 1 and throughout the manuscript.

COMMENT 4:

Figure 1 A: The quality of the image displaying the HITChip-Oligo profile data is very poor. Can a higher-resolution image be included? It may also be helpful if this image was expanded and displayed as a single image in the Supplementary Information.

Response: As suggested, we have included a high-resolution HITChip-Oligo profile image in Figure 1 (Fig. 1 A), but as a combined picture resolution is still not improved. Therefore, we have provided an expanded image of the HITChip-Oligo profile as Supplementary Figure S1 in the Supplementary Information section (Page 8, Line 176).

COMMENT 5:

Figure 2: This figure seems to be constructed using a smaller number of data points to Figure 1. Is this correct and if so explain why in the text of the manuscript. What impact does this have in the overall data analysis and conclusions?

Response:

Yes, it is correct. Figure 2 (A & B) has been constructed with the LC-HRMS metabolite profile of 71 subjects' sample data at a one-time point as baseline (summer). We followed the same baseline (summer) sample data for comparing the Indian gut microbiota with European (please see Fig. 3). Our interest here is to show the chemical profile difference between the two gut microbiota clusters (Cluster-P and Cluster-B/R, defined based on the gut microbiota community

structure, please see Fig. 1) in the study population. This sample data covers 41 samples of Cluster-P and 30 samples of Cluster-B/R. This limitation in the number of samples analysed is due to the very expensive LC-HRMS spectral library analysis. However, for the short-chain fatty profiling by HPLC, we generated data for all three seasonal samples. We mentioned this sample limitation on Page 10, Lines 232-235.

COMMENT 6:

Figure 2B and Table S3: 22 faecal metabolites were used to construct Figure 2B whereas 23 faecal metabolites that differ significantly are listed in Table S3. The additional metabolites in Table S3 appears to be compound ID C1806. If 23 compounds should be included in the analysis, correct in the manuscript and describe any subsequent impact on the conclusions from the data analysis. If 22 compounds only should be included in the analysis explain to the Editor why 23 are included in Table S3 and correct Table S3 accordingly.

Response:

Thank you for pointing out the missing data in Figure 2B. As mentioned in Supplementary Table S6 (in revision), a total of 23 compounds significantly differed between Cluster-P and Cluster-B/R (log 2 fold change and $P < 0.05$ Wilcoxon test, BH corrected). While drawing Figure 2B, a cutoff of more than log 2 fold change and $P < 0.05$ Wilcoxon test, BH corrected were included, thereby the compound ID C1806 with 2.0 log fold change missed out. As the Randomforest analysis also resulted in C1806 as a key differentiating metabolite between Cluster-P and Cluster-B/R, and the Wilcoxon test, BH corrected p-value is highly significant ($p = 7.51E-06$), C1806 is a key metabolite to consider. We have corrected Figure 2B by including the data of

C1806 with 23 faecal metabolites. As C1806 is included in Figure 2B, there is no subsequent impact on the conclusions for the data analysis.

COMMENT 7:

Lines 232-233 and Figure S1 B: Suggest that the text should read “(~20 mean decrease accuracy)” as oppose to “(> 20); as the data in Figure S1 B indicates that the Mean decrease accuracy score for C1806 is ~ 20 while that for C1767 is ~ 19.5.

Response:

Thank you for pointing out the mistake. We have corrected it as “(~20 mean decrease accuracy)” in the main text line 267, Page 13, and in the Supplementary Information Figure legend of Figure S2 (in revision) on page number 10 and line number 205. A similar correction as “(~80 mean decrease accuracy)” is also made in line number 203.

COMMENT 8:

Lines 253-258 and Figure 3A: Figure 3A is a PCA plot of European (pink dots) and Indian (blue/green dots) which separates the two groups into two clusters in particular relative to Comp. 1. However, in the text it is stated that the separation was “associated with the *Prevotella* status, irrespective of Indian or European subjects”. Please review this section to ensure there is no error, and if not, edit the text to better explain the observations from the data.

Response: Sorry for the confusing statement. We have revised the statement as “PCA using the gut bacterial compositional abundance of the most prevalent genus-like taxa (67 taxa, detected in >20% of the samples with above 2.8 log HITChip probe signal) showed two modes (Fig. 3A), separated mainly based on the *Prevotella* status in relation to the PCA component-1.

Multimodality analysis viewed a bimodality in *Prevotella melaninogenica* et rel., *Prevotella oralis* et rel., *Prevotella tanneriae* et rel., *Ruminococcus bromii* et rel., and *Streptococcus mitis* et rel. in both the nationals (India and Europe). Among the bimodality taxa, only *Prevotella* differed significantly in their abundance between Indian and European subjects.” in lines 277-284, page13.

COMMENT 9:

While the manuscript is well written, a number of minor edits for grammar and style are suggested. Many (but not all) of these relate to the Introduction. When addressed, the manuscript would benefit from a final review to specifically focusing on grammar and style.

Response: Thank you for the positive remark. We have addressed and edited all the grammatical and style errors as per the reviewer’s suggestion, particularly in the Introduction section. Please see the responses to the comments below.

COMMENT 10:

Line 63-64: Suggest moving this statement about health claims and inserting it immediately before the sentence starting on line 60 that refers to validated claims. This edit is suggested as it builds the case better to first mention “claims” followed by “validated claims”.

Response: As suggested, we moved the statement “Most health claims are on fermented foods and associated bacteria on the host metabolism and diseases.” to Lines 60-62, Page 1. Thank you for the suggestion.

COMMENT 11:

Line 66: Delete “predominantly with”

Response: Deleted “predominantly with” in Line 66, Page 1

COMMENT 12:

Line 68: Suggest editing to “of the phylum Bacillota, in addition to yeasts and fungi”

Response: Corrected as per the suggestion in line 68, Page 1.

COMMENT 13:

Line 75: Suggest editing to “whereas genetics is likely not a valid driver”

Response: Edited as per the suggestion in line 75, Page 3

COMMENT 14:

Line 78: Change “grouped” to “group”

Response: Changed as “group” in line 78, Page 3.

COMMENT 15

Line 79: Insert “and” following “Bacteroides entrotypes 1 and 2,”

Response: Inserted “and” in Line 79, Page 4

COMMENT 16

Line 80-81: Replace “the traditional population” with “human populations that follow a traditional lifestyle” and delete “the modern industrialised population” and replace with “populations exposed to modern industrialised lifestyles”

Response: We have replaced as per suggestion in Lines 80-82 on Page 4.

COMMENT 17:

Line 82: Suggest deleting “stable” at “to be stable due to”

Response: Deleted “stable” in Line 83, page 4.

COMMENT 18:

Line 84: Suggest deleting “on” and replacing with “the prevalence”

Response: Replaced with “ the prevalence of” in Line 85, Page 4

COMMENT 19:

Line 85: Suggest inserting “specific” following “individual”. Change “return” to “returning”

Response: Inserted as per suggestion in Lines 85 and 86, Page 4

COMMENT 20:

Line 89: Insert “changes in the gut microbiota populations are cyclic (annual)” Following “hunter gathers”

Response: Inserted as per suggestion in Lines 89-90, Page 4

COMMENT 21:

Line 94: Start the sentence with “Examples include” and edit remainder of sentence to ensure correct grammar.

Response: Started the sentence with “Examples include” and edited the sentence by dividing it into two to ensure correct grammar in lines 95-101, Page 4 as follows

“Examples include a symbiotic preparation with *Lactiplantibacillus plantarum* and fructooligosaccharide prevented sepsis among infants in rural India¹⁹, and *Limosilactobacillus reuteri* DSM17938 was effective in breastfed infants with colic²⁰. With the support of few human clinical studies, the International Scientific Association for Probiotics and Prebiotics (ISAPP) consensus statement on fermented foods²¹ stated evidence of health benefits in yoghurt consumption with a reduction in type-2 diabetes mellitus and cardiovascular diseases.”

COMMENT 22:

Line 103: This sentence needs to be edited with particular reference to explaining “eliminate immediately” and “shed more days”

Response:

We have edited the sentence by explaining more as follows.

“Based on these transient bacteria's sheltering ability in gut microbiota, Zhang et al.¹⁵ categorised the gut microbiota into colonisation resistance (its ability to prevent colonisation by allochthonous or exogenous bacteria by eliminating them immediately) and colonisation permissive (shelter the exogenous strain over 24–48 h).”, in Lines 108-111, Page 5.

COMMENT 23:

Line 107: Delete “by” and replace with “following consumption of”

Response: Deleted and replaced as per suggestion, in Line 114, Page 5.

COMMENT 24

Line 111: Delete “for” and replace with “using”

Response: Deleted in Line 121, page 5

COMMENT 25

Line 114: Insert “the prevalence of” before “Prevotellaceae”

Response: Inserted as suggested in line 123, page 5

COMMENT 26

Line 117: Insert “of the” following “two”

Response: Inserted as suggested in line 127, page 5.

COMMENT 27

Line 124: Replace “lives” with “living”

Response: Replaced as suggested in line 133, page 6.

COMMENT 28

Line 126: Insert “were” following “groups”

Response: Inserted as suggested in line 135, page 6

COMMENT 29

Line 127: The objective of the study is described as the faecal bacteria, while the manuscript title and the Introduction refer to “gut” bacteria. Suggest qualifying the statement by highlighting that the faecal bacteria are commonly used as a marker for the large intestine i.e. colon microbiota populations.

Response: We have changed “faecal bacteria” to “gut microbiota”, as the faecal microbiota are commonly used as a marker for the colon microbiota populations and also we assessed the Archaeal population. We have changed the sentence as follows.

“Our study intends to understand the impact of long-term consumption of fermented foods on seasonal changes in gut microbiota composition on this isolated rural Indian community by analysing the faecal samples, as the faecal microbiota are commonly used as a marker for the colon microbiota populations” in Lines 135-138, Page 6. Moreover, we have changed the “faecal bacteria” to “gut microbiota” all over the text.

COMMENT 29

Line 153: Should “and” be replaced by “or”?

Response: Replaced as suggested in Line 162, page 7.

COMMENT 30

Line 236: There is reference to acetate and propionate production but no associated data is presented. This should be highlighted as “(data not shown)”. The significance/relevance of the statement about acetate and propionate should also be included.

Response: We have included “data not shown” in Lines 253-254. The data related to acetate and propionate production is shown in Supplementary information Fig. S11, with significant changes in acetate production between diet groups, mentioned in the result section Line 490-492, Page 24. We discussed its relevance in Lines 643 -646, Page 31 in the Discussion section.

COMMENT 31

Line 223: Sentence starting with “The MS/MS” does not read well/correctly, something is missing

Response: We have edited the sentence as follows

“We assigned the identity of 23 significantly differing compounds between the two clusters by MS/MS spectral analysis by comparing it with the mzCloud library and listed in Supplementary Table S6.” Please see Lines 240-24 on Pages 10-11.

COMMENT 32

Line 399: This should read “did not differ” as oppose to “not differed” as currently in the text

Response: Corrected as per the suggestion in Line 421, Page 20

COMMENT 33

Line 445: Suggested editing “were also significantly varying over seasons” to “also varied significantly over season”

Response: Edited as per the suggestion in Line 483, page 23.

COMMENT 34

Line 548: Suggest editing to “... .. visible in traditional populations living in uncontrolled household environments,”. Also, what do the author mean by “uncontrolled”?

Response:

“Uncontrolled households” mean “ Unconditioned households”, the household environment is not air-conditioned (without proper heating or cooling or insulation facilities), and is very much exposed to outside temperature changes.

We have corrected the sentence as follows.

“The impact of seasonal weather changes on gut microbiota may be more visible in the traditional population living in an unconditioned household environment (without proper heating or cooling or insulation facilities), which is very much exposed to outside temperature changes, similar to the present study population households.” in Lines 602-606, Page 29 of the revision.

COMMENT 35

Line 555-557: Meaning/context of the statement “important concerning seasonal patterns” is unclear

Response:

Corrected the statement as follows “Our finding of seasonal changes in gut microbiota and metabolites composition seems to be important concerning the seasonal occurrence of some non-infectious diseases.” in lines 611-613, Page 29.

COMMENT 36

Line 572-573: Sentence is poorly constructed

Response:

Reconstructed the sentence as follows “The seasonal instability of gut bacteria due to long-term frequent fermented foods consumption may allow the gut more colonisation permissive and weaken the resilience nature.” in Lines 628-630, Page 30.

COMMENT 37

Line 587: Suggest editing to "... prevent pathogen establishment,"

Response: Edited in Line 644, Page 31.

COMMENT 38

Line 608: Suggest editing to "... produced it via the butyrate"

Response: Edited in Line 666, Page 32.

COMMENT 39

Line 660: HDI should be in brackets i.e., (HDI)

Response: Corrected in Line 718, Page 34

COMMENT 40

Line 694: Suggest inserting "the" immediately before "above"

Response: Inserted in Line 757, Page 35

COMMENT 41

Lines 695-696: Change "consuming" to "consumed" throughout this sentence

Response: Changed in lines 757-758, Page 35. Also, in the figure legends of Figures 5, 6, 7 and 8, Supplementary Figures S5, S6, S7, S9, S11, S12, S13, and S14.

Detailed Response to the Reviewers' Comments

Reviewer #1(Remarks to the Author):

Comment: Although I still have some concerns about the small sample size (20 subjects x group) and the obsolete methodology used, the authors have addressed all the other points raised.

Response:

We agree that the sample size is relatively small (Four diet groups, ~20 Indian subjects in each group with three seasonal samples n=214, and samples of 76 European subjects). However, it is important to note that only a very limited number of studies address the gut microbiota of people from the Mongoloid race of the isolated Meitei community in India, particularly on the impact of fermented foods consumption. In addition, an extensive screening of over 1000 subjects was needed in order to recruit ~20 subjects per group. As suggested by the Editor in the first submission (“the study cannot be strengthened with additional samples, please clearly acknowledge the small sample size as a limitation of your work”), we have acknowledged the sample size limitation and mentioned in the Discussion section concluding remarks, in line number 579-583, page 23 as follows,

“The relatively small size of samples analysed here limits the strength of our study. However, the seasonal instability in the human gut microbiota due to long-term fermented food consumption observed in our study can be taken forward for future studies with large-scale subject recruitment and sample analysis”.

Regarding the method used, we agree that shotgun metagenomics is the present-day choice for gut microbiota studies. High throughput shotgun metagenome sequencing was in the initial stages when the project was initiated (in the year 2012). Several studies have

benchmarked results obtained from the microarray (HITChip) with taxa-specific qPCR as well as high throughput sequencing assays and described a high degree of overlap between them. Therefore, the present study is comparable with those already present in the literature and for future works, and equivalent biological conclusions can be obtained.

Reviewer #2 (Remarks to the Author):

Comment 1: That authors have addressed the individual comments raised from review of the original manuscript in an adequate manner. A number of errors relating to the data or the way it was presented in the original article were acknowledged and addressed, while a number of suggested edits relating to style and grammar were amended.

Response: Thank you for recognising the corrections made as per the suggestions.

Reviewer #2 (Remarks on code availability):

Comment 2: could not find where to access code

Response: We made the source code available via Zenodo (<https://doi.org/10.5281/zenodo.14369940>) under an MIT license. The “Code Availability” is mentioned in Line 884-886, Page 35, of the final revision.